# Reshuffling of the ancestral core-eudicot genome shaped chromatin topology and epigenetic modification in *Panax*

Zhen-Hui Wang[1,2,3], Xin-Feng Wang[2], Tianyuan Lu [4], Ming-Rui Li[2], Peng Jiang[3], Jing Zhao[3], Si-Tong Liu[5], Xue-Qi Fu[5], Jonathan F. Wendel [6], Yves Van de Peer [7,8,9✉], Bao Liu [3✉] & Lin-Feng Li [2✉]

All extant core-eudicot plants share a common ancestral genome that has experienced cyclic polyploidizations and (re)diploidizations. Reshuffling of the ancestral core-eudicot genome generates abundant genomic diversity, but the role of this diversity in shaping the hierarchical genome architecture, such as chromatin topology and gene expression, remains poorly understood. Here, we assemble chromosome-level genomes of one diploid and three tetra-ploid *Panax* species and conduct in-depth comparative genomic and epigenomic analyses. We show that chromosomal interactions within each duplicated ancestral chromosome largely maintain in extant *Panax* species, albeit experiencing *ca*. 100–150 million years of evolution from a shared ancestor. Biased genetic fractionation and epigenetic regulation divergence during polyploidization/(re)diploidization processes generate remarkable biochemical diversity of secondary metabolites in the *Panax* genus. Our study provides a paleo-polyploidization perspective of how reshuffling of the ancestral core-eudicot genome leads to a highly dynamic genome and to the metabolic diversification of extant eudicot plants.

---

[1] Faculty of Agronomy, Jilin Agricultural University, 130118 Changchun, China. [2] Ministry of Education Key Laboratory for Biodiversity Science and Ecological Engineering, School of Life Sciences, Fudan University, 200438 Shanghai, China. [3] Key Laboratory of Molecular Epigenetics of the Ministry of Education (MOE), Northeast Normal University, 130024 Changchun, China. [4] McGill University and Genome Quebec Innovation Center, Montreal, QC H3A 0G1, Canada. [5] School of Life Sciences, Jilin University, 130061 Changchun, China. [6] Department of Ecology, Evolution & Organismal Biology, Iowa State University, Ames, IA 50011, USA. [7] Department of Plant Biotechnology and Bioinformatics, Ghent University and VIB Center for Plant Systems Biology, Gent, Belgium. [8] Department of Biochemistry, Genetics and Microbiology, University of Pretoria, Pretoria, South Africa. [9] College of Horticulture, Academy for Advanced Interdisciplinary Studies, Nanjing Agricultural University, 210095 Nanjing, China. ✉email: yves.vandepeer@psb.ugent.be; baoliu@nenu.edu.cn; lilinfeng@fudan.edu.cn

Polyploidy or whole genome duplication (WGD) is a ubiquitous phenomenon in angiosperms[1,2] and all extant flowering plants likely evolved from a polyploid ancestor[3]. Recurring polyploidization and (re)diploidization events in flowering plants led to highly dynamic plant genomes[4–7]. However, it remains poorly characterized how these often-cyclical genome doubling/diploidization processes have contributed to angiosperm evolution and diversification.

It is evidenced that all extant core-eudicots share an ancient WGD, usually referred to as the γ-triplication event[3,8]. The ancestral core-eudicot genome was restored to a diploid karyotype, and most of the extant core-eudicot species experienced additional paleo-polyploidization events during their independent diversification processes[1,2]. Frequent genome doubling followed by independent diploidization and chromosomal rearrangement processes have provided extant core-eudicots with structural genomic and phenotypic diversity[5,9]. An example of such genome structural evolution is cotton (*Gossypium*) where, following polyploidization, both large genomic fragment reorganization (i.e., chromosome fusion and fission) and individual gene repertoire evolution (i.e., biased genetic fractionation) have generated phenotypic novelty and species diversification[10–12]. However, relatively little is understood about how reshuffling of the duplicated ancestral core-eudicot genome affected genome plasticity of extant core-eudicot plants and the underlying mechanisms responsible for the genetic and epigenetic partitioning of a duplicated genome. The genus *Panax* (Araliaceae) includes four diploids, three tetraploids and one species complex[13]. It has been shown that this genus shares the core-eudicot γ triplication and has undergone two additional *Panax*-specific WGDs (Pg-β and Pg-α)[14,15]. In particular, as a medically important genus, all ginseng species contain a large number of secondary metabolites[16,17]. These attributes make the ginseng genus an ideal system to elucidate how the reshuffling of the duplicated (or better triplicated) ancestral core-eudicot genome has affected genome structure, epigenetic regulation and secondary metabolites diversity of extant plant species after repeated polyploidization and (re)diploidization events.

In this study, we assemble chromosomal-level reference genomes of one diploid (experienced γ and Pg-β) and three closely related tetraploid (experienced γ, Pg-β and an additional, more recent, Pg-α duplication) *Panax* species. We infer the evolutionary history of the seven ancestral core-eudicot chromosomes (Eu1-Eu7) in the four *Panax* species. Based on this paleo-polyploidization framework of the genus *Panax*, our genome-wide comparisons of the three-dimensional (3D) genome architecture and cytosine methylation and gene expression dynamics (mRNA, lncRNA and small RNA) further reveal that reorganization of the ancestral genome structure is associated with the reconfirmation of chromatin topology and epigenetic regulation divergence in extant *Panax* genomes. Our study thus provides a genome-wide landscape view of how polyploidization and subsequent (re)diploidization contribute to genome structure plasticity and metabolomic diversity of extant *Panax* species.

## Results

**Genome assembly, gene annotation and quality control.** Our chromosome analyses confirmed the diploid ($2n = 2x = 24$) and tetraploid ($2n = 4x = 48$) karyotypes of the four *Panax* species (Supplementary Fig. 1). Genome sizes of the four species were estimated by genome survey (Table 1 and Supplementary Fig. 2) and flow cytometry (Supplementary Fig. 3), respectively. To obtain reliable inference of the karyotype evolution, we employed three different strategies to de novo assemble the four *Panax* species (Supplementary Note 1–2). The resulting assemblies were 1.96 Gb and 2.02 Gb for *P. stipuleanatus* and *P. japonicus*, with a contig N50 of 2.88 Mb and 1.58 Mb for the two species, respectively (Table 1). Total lengths of the assembled genomes were relatively larger for *P. ginseng* (3.36 Gb) and *P. quinquefolius* (3.57 Gb), with a contig N50 of 19.75 Mb and 0.87 Mb for the two species, respectively. Genome annotation of the four *Panax* species identified 41,224–74,307 protein-coding genes (Table 1).

Assessments of the genome quality revealed high gene completeness (BUSCO = 93.00–95.14%) and genome contiguity (LAI = 7.13–16.24) for all four species (Table 1). In particular, genome contiguity measures of *P. stipuleanatus* (LAI = 12.85) and *P. japonicus* (LAI = 16.24) were comparable to the model species *Arabidopsis thaliana* (LAI = 15.62) and *Vitis vinifera* (LAI = 14.58)[18,19], although the two *Panax* species had much larger genome sizes. Genome collinearity analyses showed that, while the four species varied dramatically in genome size, they still maintained high collinearity across the 12 orthologous chromosomes (Supplementary Fig. 4). Based on the genome collinearity and sequence homoeology to diploid relatives, we further separated the 24 chromosomes of the three tetraploid species as two subgenomes (Supplementary Table 1). Phylogenetic inference based on orthologous genes revealed that subgenome B of the three tetraploid species clustered with the diploid *P. notoginseng* while subgenomes A formed a monophyletic clade (Supplementary Fig. 5). These genomic features together corroborated the quality of the genome assemblies of the four *Panax* species.

**Reconstruction of the ancestral karyotype in the modern *Panax* genome.** The polyploidization history of the four *Panax* species was estimated by calculating synonymous substitution rates (Ks) between homologous gene pairs. Our results confirmed that the genus *Panax* experienced the core-eudicot-shared γ triplication and two additional lineage-specific duplications (Pg-α and Pg-β)[14,15,20] (Supplementary Fig. 6). Likewise, we also identified the other previously inferred paleo-polyploidizations (i.e., Dc-β) in *Daucus carota* (carrot) and *Lactuca sativa* (lettuce)[21]. Karyotype evolution of the ancestral core-eudicot genome in extant *Panax* species was inferred by analyzing genome collinearity between grape (putative post-γ core-eudicot genome) and the above selected six species (extant core-eudicot genomes). Our genome-wide comparisons identified more collinear orthologous genes (referred to as ancestral genes) in the four *Panax* species (16,010–31,729) than those of carrot (13,985) and lettuce (14,087) genomes (Supplementary Table 2). In particular, these ancestral genes tend to be retained in *Panax* genomes as large contiguous genomic blocks, i.e., on average about 17–20 (95% confidence interval (CI)) ancestral genes localized in each of these collinear genomic blocks (Supplementary Figs. 7–16 and Supplementary Table 2). In contrast, a smaller number of collinear ancestral genes have been retained in the carrot (95% CI: 12–13) and lettuce (95% CI: 11–12) genomes (t-test, p value < 0.01). Together, our results indicate that the ancestral core-eudicot genome has been well-preserved in modern *Panax* genomes, even after several rounds of polyploidization-diploidization.

Based on genome collinearity analyses, we identified 26 post-γ and four post-Pg-β chromosomal fusion events in the *Panax* genomes, 10 and 20 of the post-γ events were also characterized in lettuce and carrot genomes, respectively (Supplementary Figs. 8–16). Given the shared ancestral (γ) and lineage-specific (i.e., Pg-α and Pg-β) polyploidization/(re)diploidization histories of the six extant core-eudicot species[20–22], we propose an evolutionary framework wherein the 21 (after hexaploidy/ triplication) post-γ ancestral core-eudicot chromosomes (A1-A7, B1-B7 and C1-C7) were rearranged into eight pre-Pg-β ancestral chromosomes (Ar1-Ar8) through the identified 26 post-γ chromosomal fusions (Fig. 1 and

**Table 1 Statistics of genome features of the four *Panax* species.**

| Genome information | P. stipuleanatus | P. japonicus | P. ginseng | P. quinquefolius |
|---|---|---|---|---|
| Genome size (Gb)[a] | 2.15 | 2.09 | 3.41 | 3.60 |
| Total length of contigs (Gb) | 1.96 | 2.02 | 3.36 | 3.56 |
| GC content (%) | 35.24 | 33.93 | 34.25 | 34.11 |
| N50 length (contig) (Mb) | 2.88 | 1.22 | 19.75 | 0.87 |
| Predicted genes | 41,224 | 74,307 | 65,913 | 64,247 |
| Average transcript length (bp) | 1280 | 1267 | 1389 | 1394 |
| Average CDS length (bp) | 1037 | 1061 | 1119 | 1149 |
| Average exon length (bp) | 273 | 265 | 270 | 264 |
| Average intron length (bp) | 925 | 916 | 835 | 891 |
| BUSCO (%) | 93.10 | 93.00 | 95.14 | 93.93 |
| LTR Assembly Index (LAI) | 12.85 | 16.24 | 7.13 | 7.99 |

[a]Genome size was estimated by genome survey.

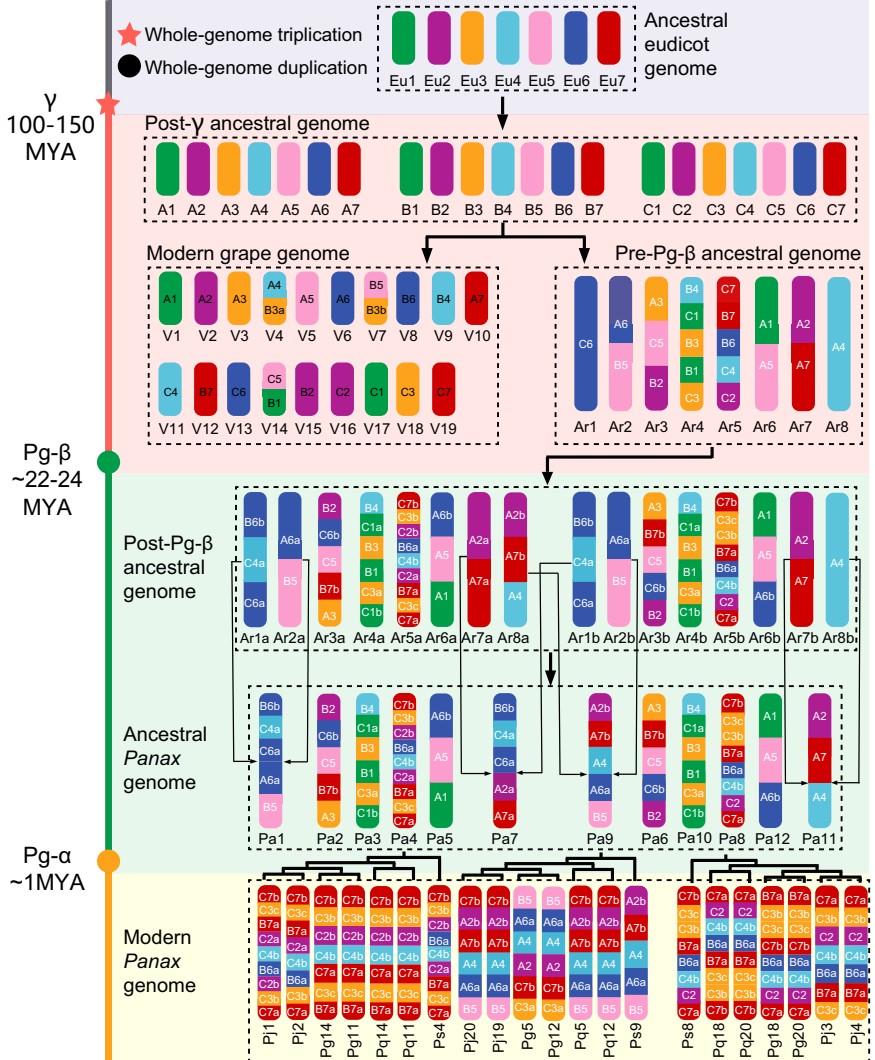

**Fig. 1 Evolutionary rearrangements of the ancestral core-eudicot genome to generate the genome of extant ginseng species.** The ancestral core-eudicot genome is hypothesized to contain seven chromosomes (Eu1–Eu7) marked with seven different colors. The post-γ ancestral genome consists of three chromosomal compartments (A1–A7, B1–B7 and C1–C7) which were reunited into the same nucleus some 100–150 million years ago (MYA). The extant grape genome (V1–V19) has evolved from the post-γ karyotype through one chromosomal fission and two fusions. In parallel, the post-γ ancestral genome was structured into a pre-Pg-β karyotype with eight chromosomes (Ar1–Ar8). The pre-Pg-β karyotype was doubled about 22–24 MYA (Ar1a–Ar8a and Ar1b–Ar8b) but further reorganized into the ancestral *Panax* genome with 12 chromosomes (Pa1–Pa12). After the Pg-α duplication, one fragment conversion and two translocations occurred in the four extant *Panax* species (Ps, *Panax stipuleanatus*; Pj, *Panax japonicus*; Pg, *Panax ginseng*; Pq, *Panax quinquefolius*). Red star, green and orange circles present the γ triplication, Pg-β and Pg-α duplication, respectively.

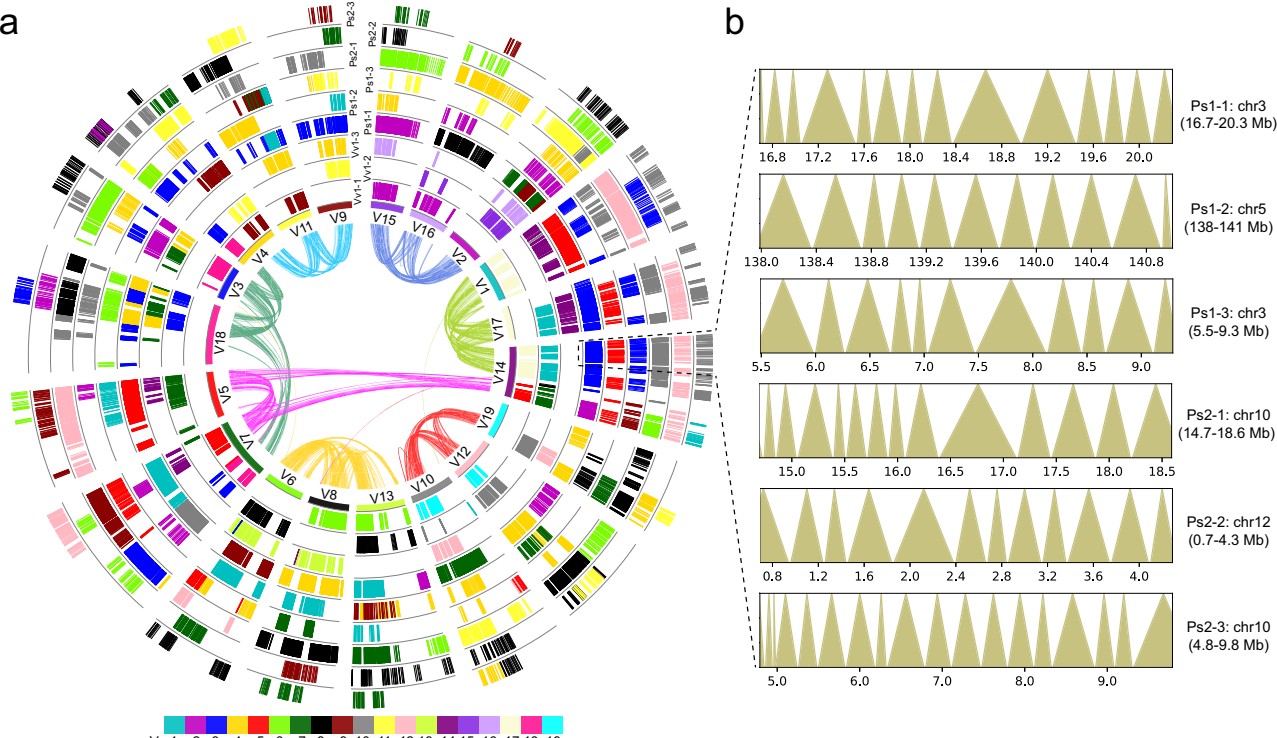

**Fig. 2 Global alignment of ginseng genomic regions to the grape genome and chromatin topology of the paleo-polyploidization-derived homologous regions in ginseng. a** Collinearity was determined by genes (referred to as ancestral core-eudicot gene) that were collinear between ginseng and grape genomes. The 19 extant grape chromosomes (V1–V19) in the innermost circle are color-coded to different colors according to the color bar at the bottom. Curved lines within the inner circle connect homoeologous genes duplicated by the γ triplication. Colors of these curved lines correspond to the seven ancestral core-eudicot chromosomes (Eu1–Eu7)[19]. A genomic region in grape has three and six homologous regions compared to itself and ginseng genomes, respectively. The three grape circles (Vv1-1, Vv1-2, and Vv1-3) are the result of the core-eudicot γ triplication. The middle three (Ps1-1, Ps1-2, and Ps1-3) and outside three (Ps2-1, Ps2-2, and Ps2-3) circles are also the result of the core-eudicot shared γ triplication plus the Pg-β duplication, respectively. The short lines within each genomic region of the nine circles represented the predicted ancestral core-eudicot genes. Grape genes have initials "Vv" or "V" and ginseng genes "Ps". Colors of these core-eudicot genes represent their physical locations on the 19 grape and 12 ginseng chromosomes, respectively. The six homologous genomic regions included in black dashed line were selected to compare chromatin topology. **b** From top to bottom are the topological association domain-like (TADs-like) structures of the six selected homologous genomic regions. Both the top three (extant *Panax* chromosome 3, 3, and 5) and bottom three (extant *Panax* chromosome 10, 10, and 12) regions resulted from the γ triplication, while the top and bottom three homologous genomic regions, in turn, resulted from the Pg-β duplication (giving rise to Ps1 and Ps2). The numbers on x-axis indicate physical positions on extant ginseng chromosomal segments. Triangle size represents the length of each TAD on the ginseng chromosome. Source data are provided as a Source Data file.

Supplementary Figs. 8–16). Thereafter, the post-Pg-β genome (Ar1a-Ar8a and Ar1b-Ar8b) was structured into the ancestral *Panax* genome (Pa1-Pa12) via four post-Pg-β chromosomal fusion events (Fig. 1). Among the extant *Panax* genomes, we also identified three chromosomal rearrangements, including one fragmental inversion on *P. stipuleanatus* chromosome 4, one reciprocal translocation between *P. stipuleanatus* chromosomes 8 and 9, and one inversion on *P. notoginseng* chromosome 6 (Fig. 1 and Supplementary Figs. 17 and 18).

The above inferences have revealed the evolutionary transformation of the seven ancestral core-eudicot chromosomes (Eu1–Eu7) into 42 homoeologous genomic regions (referred as to duplicated ancestral core-eudicot chromosome) in extant *Panax* genomes after the γ triplication (3×) and Pg-β duplication (2×) (Fig. 1). We then allocated all identified collinear genes to the 42 ancestral core-eudicot chromosomes (Fig. 2a and Supplementary Data 1). In the *P. stipuleanatus* genome, for example, biased genetic fractionation of the gene duplicates was a general phenomenon in all the 42 ancestral core-eudicot chromosomes, with only 993 (6.7% of total) of the ancestral genes retaining more than half (>3) of the duplicate pairs (Supplementary Data 1). In addition, our comparisons also showed that genes duplicated by the more recent Pg-β duplication (6484 gene pairs) were less fractionated compared to those derived

from the more ancient γ triplication (4143 gene pairs). Further genome-wide comparisons of the fractionation pattern confirmed that gene duplicates derived from the same ancestral gene showed different retention rates along the ancestral core-eudicot chromosomes (Supplementary Fig. 19). For example, even though the six homologous genomic regions (marked with purple color in Fig. 1) in the extant *P. stipuleanatus* genome (chromosomes 2, 4, 6, 7, 8, and 11) were duplicated from the same ancestral core-eudicot chromosome Eu2, the numbers of retained ancestral genes differed dramatically along the γ-derived triplicates (i.e., among the Ps1-1/Ps1-2/Ps1-3 or Ps2-1/Ps2-2/Ps2-3) (Supplementary Fig. 19 and Supplementary Data 2). In contrast, Pg-β-derived gene duplicates (i.e., between Ps1-1 and Ps1-2, Ps1-2 and Ps2-2 or Ps1-3 and Ps2-3) showed similar gene fractionation rates along the ancestral core-eudicot chromosomes.

We next focused on how these ancestral genes duplicated in the different WGDs evolved in the diversification process of extant *Panax* species. Our pan-genomic analyses assigned these ancestral genes to 29,499 orthogroups, only 1836 (6.2% of the total) of which were specific to each of the seven extant *Panax* genomes (one diploid and six tetraploid genomes) (Supplementary Table 3). Further collinearity comparisons revealed that 6874 (32.6–49.4%) of these ancestral genes have been retained in the seven extant *Panax*

genomes as collinear orthologous genes (Supplementary Table 4 and Supplementary Data 3). Among the three tetraploid species, we identified 13,679 (52.1% of the total) and 14,550 (57.6%) collinear orthologous genes in the subgenomes A and B, respectively (Supplementary Table 5 and Supplementary Data 4). It is notable that while the three tetraploid species showed high genome collinearity (see Supplementary Fig. 4), *P. japonicus* possesses a substantially smaller genome (2.02 Gb) compared to *P. ginseng* (3.36 Gb) and *P. quinquefolius* (3.57 Gb) (see Table 1). This phenomenon can be explained, at least partially, by the different evolutionary history of long terminal repeats (LTRs). For example, compared to *P. ginseng*, the increased genome size of *P. quinquefolius* was likely due to the post-speciation (<1 MYA) burst of unknown LTRs (Supplementary Figs. 12–13). In contrast, while *P. japonicus* experienced the shared pre-speciation LTR burst, the majority of the *Copia*, *Gypsy* and other unknown retrotransposons have expanded more recently. Distinct expansion patterns of the three retrotransposon families were also observed for the two diploid species, *P. stipuleanatus* and *P. notoginseng* (Supplementary Figs. 20–21). Together, these features suggest that biased fractionation, together with broad-scale chromosomal rearrangements have resulted in extensive diversification of genome structure in extant *Panax* species.

**Association between core-eudicot genome repatterning and chromatin topology.** The evolutionary rearrangement of chromosomes and genome content can profoundly affect chromatin topologies[23]. Evidence from cotton and other plant species confirmed that polyploidization reshapes the chromatin topology of newly formed polyploid genomes[24,25]. However, it is largely unknown how the reshuffling of the ancestral core-eudicot genome affected the remodeling of chromatin topology in extant eudicot genomes. Based on the above described paleo-genomic framework, we investigated whether the chromatin topology observed in the extant *P. stipuleanatus* genome was associated with the evolutionary history of the ancestral core-eudicot genome. We assume that, if all the duplicated ancestral core-eudicot chromosomes were fully preserved, the 42 homologous genomic regions in extant *Panax* genomes would have maintained similar chromatin topologies. However, our inference of the ancestral core-eudicot karyotype evolution revealed considerable DNA-based genomic structural variation in the extant *Panax* species. Therefore, we wondered whether the chromatin topologies of duplicated ancestral chromosomes were randomly reestablished in the extant *P. stipuleanatus* genome. To this end, we studied the 3D genome architecture of *P. stipuleanatus* at the chromosome level. Genome-wide comparisons of chromatin interactions at 100 Kb resolution identified 12 interaction blocks corresponding to the *Panax* chromosomes (Supplementary Fig. 22a). This observed higher level of intra-chromosomal interactions compared to inter-chromosomal interactions (*t*-test, *p* < 0.01) indicates remodeling of chromatin topology in the *Panax* genome during the polyploidization/(re)diploidization processes.

To further examine this phenomenon, we compared the chromatin topologies between the homologous genomic regions derived from γ and Pg-β based on the same interaction matrix. Our analyses revealed that the post-Pg-β chromosome pairs (Ar1a-Ar8a and Ar1b-Ar8b in Fig. 1) showed no significantly higher values of the chromosomal interaction (estimated by the log2-normalized frequencies of the valid read pairs among the different genomic regions) compared to the other duplicated chromosomes (*t*-test, *p* = 0.45). For example, the overall chromatin interaction between the chromosomes Chr2 (Ar3a) and Chr6 (Ar3b) (from −7.083 to −8.808) was similar to the other inter-chromosomal comparisons (i.e., Chr2 vs. Chr3 (Ar4a) and Chr6 vs. Chr3) (from −7.236 to −9.051) (Supplementary

Fig. 22b). Similarly, most of the eight post-Pg-β chromosome pairs also differ in the activated (A)/inactivated (B) chromatin compartments (Supplementary Figs. 23–25 and Supplementary Data 5). This trend was more evident in the distribution pattern of sub-megabase topologically associating domain-like (TAD-like) structures, where the total number and length of TAD-like structures varied dramatically between the post-Pg-β chromosome pairs (Fig. 2b, Supplementary Figs. 26–28 and Supplementary Data 6). Nevertheless, the epigenetic modification patterns of TAD-like structures were broadly consistent with previous observations[26,27], with the TAD-like regions showing hypermethylation at cytosine sites and lower levels of gene expression compared to the border regions (Supplementary Fig. 29). These features indicate that chromatin topology remodeling of the duplicated ancestral core-eudicot chromosomes has further increased the 3-D genome diversity of the extant *Panax* species.

It is notable that the degree of intra-chromosomal interaction was broadly consistent with the reorganization patterns inferred from comparison to the ancestral core-eudicot chromosomes (Eu1–Eu7) (Fig. 3a, b and Supplementary Figs. 23–25). A general pattern was that chromatin interactions between the homologous genomic regions derived from the same ancestral core-eudicot chromosome were stronger than those between genomic regions from different ancestral chromosomes. For example, both the extant *P. stipuleanatus* chromosomes 2 and 6 are homologous to the post-γ ancestral chromosomes B2 (purple frame) and C6b (blue frame) (Fig. 3b). Levels of the chromatin interaction within the two segments were significantly stronger compared to those between the two segments and the other genomic regions (*t*-test, *p* < 0.01). In line with this, we also observed that the A/B compartment switching genomic regions broadly overlapped with ancestral chromosome fusion/fission sites (Fig. 3a, b and Supplementary Figs. 23–25). Nevertheless, we did not find a similar correlation between the TAD-like structure and the ancestral core-eudicot karyotype (Fig. 3c), possibly due to localized regional DNA sequence divergence. Together, these results suggest that while the chromatin topologies (compartment A/B and TAD-like) of duplicated ancestral core-eudicot chromosomes were reestablished in extant ginseng genome, their intra-chromosomal interactions have been largely maintained during the polyploidization/(re)diploidization processes.

**Epigenetic regulation divergence of the duplicated ancestral genes.** Reorganization of the ancestral core-eudicot chromosomes resulted in a nested pattern of duplicated genes and genomic regions and an altered chromatin topology. We then examined whether this repatterning of the ancestral core-eudicot genome has also promoted epigenetic regulation divergence of gene duplicates in extant *Panax* genomes. By comparing the patterns of gene expression and cytosine methylation, we found that, after the large-scale reorganization of duplicated ancestral chromosomes, a large proportion of the retained gene duplicates showed tissue-biased expression (39.7% of the total) (Supplementary Data 7) and differential cytosine methylation (21.1%) (Supplementary Data 1). Taking the Pg-β duplicate segment C6b as an example, the above comparisons revealed remodeling of the chromatin topologies of the two homologous regions in the extant *Panax* genome (see Fig. 3a–c). Here, our biased fractionation analyses further confirmed that only 69 (24.7% of the total) Pg-β-derived genes retained both duplicate pairs (Supplementary Data 7). In contrast, 98 (35.1%) and 112 (40.1%) ancestral genes evolved back to singleton status (i.e., lost their duplicate again) on each of the two C6b duplicated segments. Both the singleton genes as well as the retained duplicates differed in patterns of gene expression and cytosine methylation (Fig. 3d, e), which indicates that epigenetic

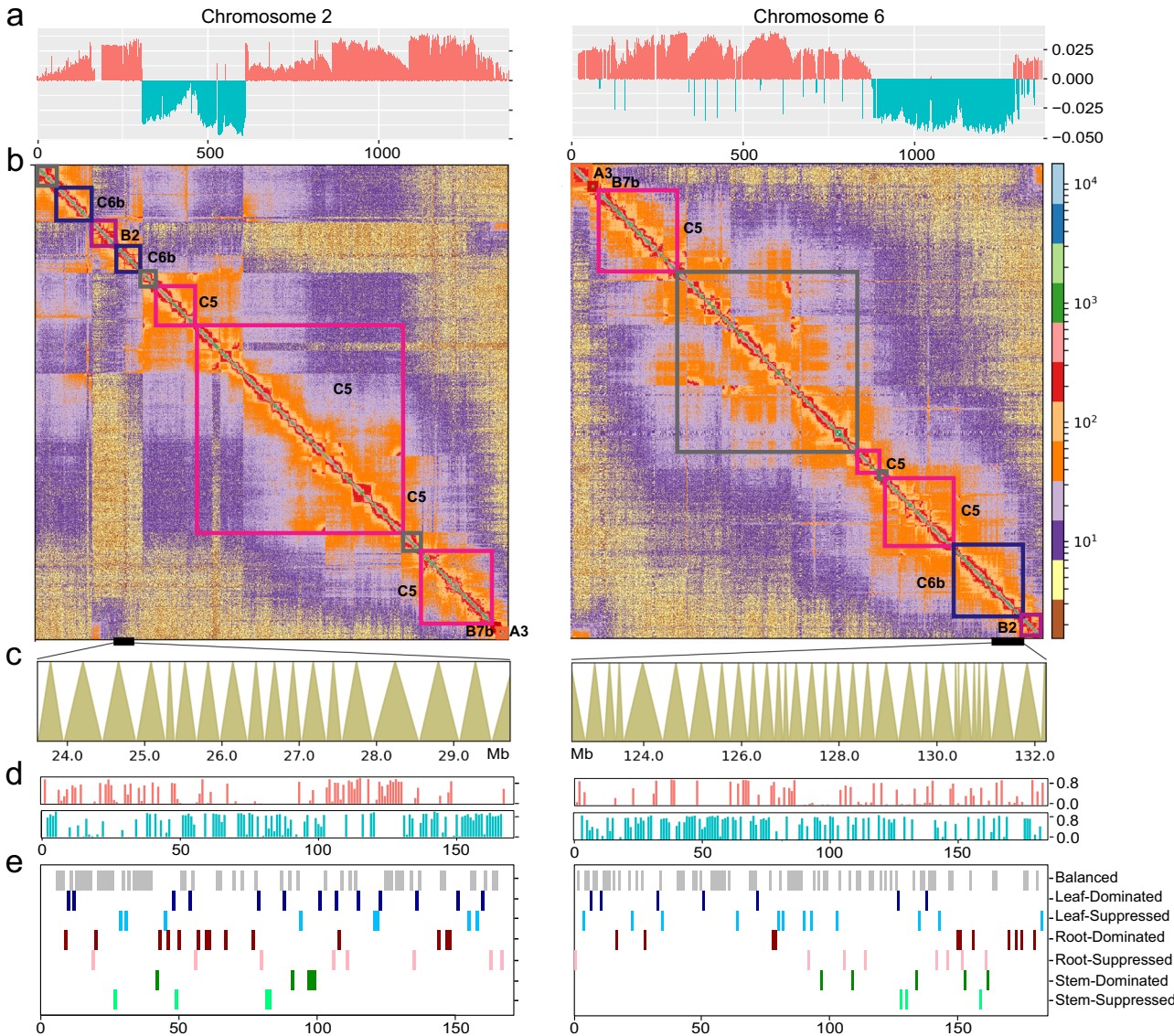

**Fig. 3 Three-dimensional (3-D) genome architecture, cytosine methylation and gene expression patterns in extant *Panax stipuleanatus* genome.**
**a** Activated A (red) and inactivated B (blue) compartments at 100 Kb resolution on extant *Panax* chromosomes 2 and 6. The two chromosomes were duplicated through the Pg-β duplication event. *X*- and *Y*-axis are the numbers of 100 Kb sliding bins and PCA eigenvectors of A/B compartments, respectively. Coordinates from left to right on chromosome 2 mirrored those on chromosome 6. **b** Heatmap of the chromatin interaction map at 100 Kb resolution. The color scheme on the right indicates the levels of chromatin interaction between the 100 Kb sliding bins. Colored boxes within the heatmap represent the ancestral core-eudicot chromosomes. Colors and names of these ancestral core-eudicot chromosomes are the same as in Fig. 1. The gray color box represents the homologous genomic region that was lost in extant grape genome. **c** Distribution pattern of topological association domain-like (TAD-like) structures on the post-γ segment C6b. The C6b segment on extant chromosomes 2 and 6 were duplicated by Pg-β. The *x*-axis indicates the physical position on extant *P. stipuleanatus* chromosomes. Each triangular shape represents a TAD-like structure. **d** Cytosine methylation of the duplicated ancestral core-eudicot genes on the two homologous segments C6b derived from Pg-β. Red and blue lines are, respectively, singletons and retained duplicates in the two segments C6b. *X*- and *Y*-axis denote the number of ancestral genes and methylation level, respectively. **e** Expression patterns of the duplicated ancestral core-eudicot genes in the two segments C6b. *X*-axis is the number of ancestral core-eudicot genes. Colors of these expressed genes indicate the seven expression patterns. Source data are provided as a Source Data file.

regulation divergence of the duplicated ancestral genes has promoted the epigenetic regulation divergence of extant *Panax* species.

We next addressed how these ancestral genes duplicated by distinct WGDs interact in the highly plastic *Panax* genome. Our analyses of the gene co-expression network revealed that the retained genes derived from the seven ancestral core-eudicot chromosomes exhibited similar degrees of functional connection in extant *Panax* genomes (Supplementary Fig. 30 and Supplementary Table 6). In the leaf tissue, for example, while the total numbers of genes identified in the leaf-related regulatory module varied among

the seven ancestral core-eudicot chromosomes (from 217 to 388), functionally important genes showed nearly equal contributions to the leaf development processes (*t*-test, all *p* values > 0.01), i.e., photosynthesis, kinase and synthase (Fig. 4a, Supplementary Fig. 30 and Supplementary Table 7). A similar phenomenon was also observed in the overall epigenetic regulation dynamics, where the duplicated ancestral core-eudicot chromosomes did not show dramatic changes in patterns of gene expression and cytosine methylation in extant *Panax* genomes (Fig. 4b, c and Supplementary Fig. 31).

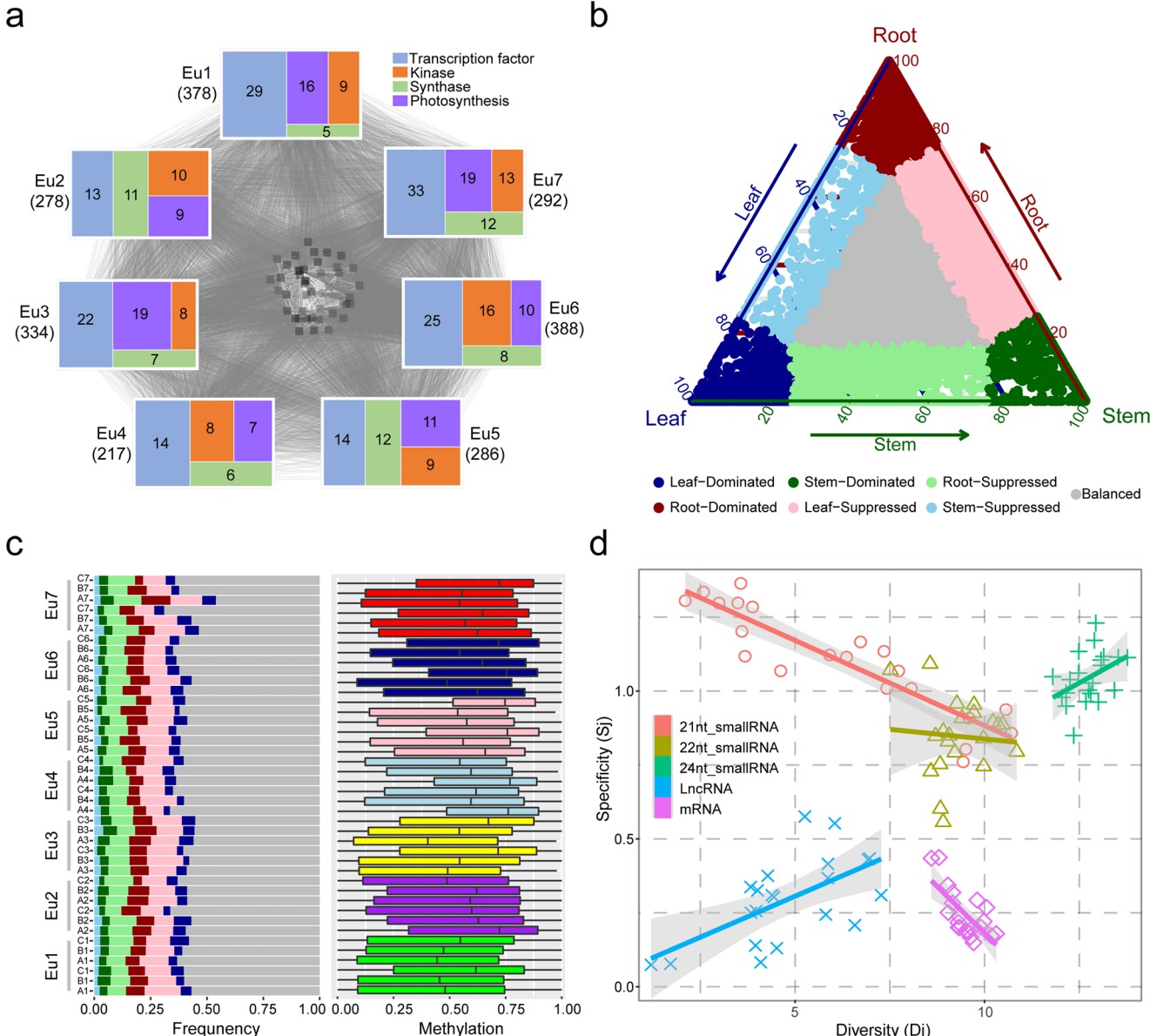

**Fig. 4 Evolutionary dynamics of the ancestral core-eudicot genes in extant *Panax stipuleanatus* genome. a** Co-expression network of the ancestral core-eudicot genes involved in the leaf development in extant *P. stipuleanatus* species. Eight gene clusters were defined according to their originations. Eu1–Eu7 are the seven ancestral core-eudicot chromosomes. Numbers below each ancestral chromosome (Eu1–Eu7) are the total identified ancestral genes. Gray dots in the middle are the ancestral genes that cannot be unassigned to the seven ancestral chromosomes. Gray lines indicate the ancestral gene interaction. Colors and numbers represented the functions and numbers of ancestral genes involved in the leaf development. **b** Ternary plot of expression patterns of ancestral genes in extant *P. stipuleanatus* genome. Each circle is a triad showing relative expression abundance in leaf, root and stem tissues for each ancestral gene. Triads in vertices correspond to the three tissue-dominant expression categories, whereas triads close to edges and between vertices correspond to three tissue-suppressed expression categories. Balanced triads are shown in gray. **c** Percentage of triads in each category of the ancestral genes (left) and boxplot of cytosine methylation for the ancestral genes (right). The 42 bars from top to bottom represent the duplicated ancestral core-eudicot chromosomes (Eu1–Eu7). Each ancestral chromosome contains six homologous genomic regions in extant *P. stipuleanatus* genome. Colors of the seven types of genes and names of each ancestral chromosome are the same as in Fig. 1. All ancestral genes identified in the 42 genomic regions were used to the statistical analyses. In right panel, the lower and upper whiskers of each colored box are the lowest (0.00) and highest (1.00) cytosine methylation level, respectively. The black solid lines within each box are media values. Lengths of the boxes are the interquartile range. **d** Expression dynamics of the protein-coding genes and non-coding RNAs of the 42 duplicated ancestral chromosomes in the leaf, root, and stem tissues. X- and Y-axis are expression diversity (Dj) and specificity (Sj) of the ancestral genes. Each symbol represents an ancestral chromosome. Differences in Dj and Sj values indicate high expression diversity among the ancestral chromosomes and expression specificity among the three tissues. The edges of each shade area are the error bands. Source data are provided as a Source Data file.

The above observations indicate that while the genomic regions duplicated from the same ancestral core-eudicot chromosome (Eu1–Eu7) showed dramatic biased genetic fractionation and divergence in epigenetic regulation, genes retained on each of the duplicated genomic regions had similar functional contributions to the tissue development of extant *Panax* species. We then examined whether the genes retained within these duplicated ancestral core-eudicot chromosomes were involved in similar

molecular functions after the repeated polyploidizations/(re) diploidizations. As expected, biased fractionation has resulted in a complementary retention pattern of the duplicated ancestral genes, i.e., only 6.7% of ancestral genes have retained all copies of their duplicates (see in Supplementary Data 1). In particular, the retained genes on each of the duplicated ancestral core-eudicot chromosome are involved in similar KEGG pathways, i.e., 62.6–78.6% of the KEGG pathways shared in more than half (>3) of the duplicated ancestral chromosomes (Supplementary Fig. 32). Likewise, the majority of the balanced-expression genes also shared similar molecular functions among the duplicated ancestral core-eudicot chromosomes (Supplementary Fig. 33). In contrast, tissue-dominant or -suppressed genes showed functional divergence among the duplicated ancestral core-eudicot chromosomes. This phenomenon is associated with gene functions, where balanced-expression genes were mainly enriched in basic cellular activities (i.e., TCA cycle, mRNA surveillance and ubiquitin mediated proteolysis), but tissue-dominant or -suppressed genes were functionally related to environmental adaptations (i.e., photosynthesis and nitrogen metabolism) (Supplementary Fig. 34). These observations suggest that the complementary retention of gene duplicates may have – at least partly – dealt with functional redundancy in extant *Panax* species.

It is notable that the regulatory non-coding RNAs (lncRNAs and small RNAs) exhibited relatively higher expression divergence than protein-coding genes among the duplicated ancestral core-eudicot chromosomes (Fig. 4d and Supplementary Fig. 35). Non-coding RNAs are RNA molecules transcribed from the genome but not translated into proteins[28]. Both the lncRNAs and small RNAs play crucial regulatory roles in a variety of biological processes by modulating gene expression at the transcriptional and post-transcriptional levels[29]. Here, the observed high expression divergence of non-coding RNAs suggest that DNA-based structural reorganization may have impacted birth-death (expressed-silent) of these RNA molecules among the duplicated ancestral core-eudicot chromosomes. Together, our findings suggest that, while the evolutionary dynamics of individual ancestral genes varied dramatically at both genetic and epigenetic levels, overall patterns of the molecular function and epigenetic regulation remained relatively stable among the duplicated eudicot ancestral core-eudicot chromosomes in extant *Panax* species.

### Evolutionary contributions of the duplicated ancestral core-eudicot genes to metabolomic diversity.

The evolutionary role of paleo-polyploidization in genome evolution and phenotypic diversification of angiosperms has long been of interest[1,30–32]. Our comparative analyses revealed that the repeated polyploidization/(re)diploidization processes have resulted in high dynamics of genome structure and epigenetic regulation of extant *Panax* species. Here, we further investigated whether this reshuffling of the ancestral core-eudicot genome has also promoted phenotypic diversification. Our results showed that, while the post-polyploidization genome contraction is observed at both the chromosomal and individual gene levels, gene families related to secondary metabolites were significantly expanded in the *Panax* genus and other selected eudicot species, especially those involved in the phenylpropanoid, sesquiterpenoid and triterpenoid biosynthesis pathways (Supplementary Figs. 36–38). Plant secondary metabolites are low molecular weight organic compounds, which not only function as signal molecules to regulate plant growth and development, but also mediate interactions with various biotic and abiotic stresses[33]. In eudicots, the majority of secondary metabolites, such as terpenoids, steroids and cyanogenic glycoside, are catalyzed by the cytochrome P450 (CYP) superfamily[33]. We explored the evolutionary roles of polyploidization-derived CYPs in the diversification of plant secondary metabolites.

As the largest family of enzymes in plant metabolism, all CYPs in angiosperms were derived from 11 ancestral genes with variable patterns of post-polyploidy retention and additional duplication[34]. Here, we identified candidate genes of nine major *Arabidopsis* CYP clades in *Panax* and other representative eudicot species (Supplementary Fig. 39). As expected, these extant core-eudicot species possessed distinct copy numbers of the nine CYP clades (Supplementary Fig. 39). In the ginseng genus, for example, highly variable copy numbers of the CYPs among the WGD-derived genomic regions (i.e., post-Pg-β chr2 and chr6) are possibly due to the independent retention of duplicated CYPs during the polyploidization/(re)diploidization processes (Supplementary Fig. 40 and Supplementary Data 8). At the epigenetic regulation level, compared to the total number of genes that showed hypermethylation (mean: 57.1% (95% CI: 49.6–50.1%)) and balanced expression (60.3%) (Supplementary Data 7), the CYPs were preferentially hypomethylated (mean: 24.0% (95% CI: 20.1–27.9%)) and exhibited tissue-biased expression (79.3%) (Supplementary Data 9), which suggests that biased genetic fractionation and divergent epigenetic regulation of the duplicated CYPs may have promoted the diversification of secondary metabolites in extant *Panax* species.

We next focused specifically on the ginsenosides, which are the major triterpene saponin and found almost exclusively in *Panax* species[16,35]. Triterpene saponins are one of the largest and most structurally diverse plant-specialized metabolites, which play important roles in, for example, plant antifungal and antibacterial activities[36–38]. In eudicots, the subfamily CYP716 (belonging to CYP85 clade) is a major contributor to the diversification of triterpenoid biosynthesis[39]. Through analyzing the paleo-polyploidization history of the CYP716 subfamily, our results showed that the ginseng genus contained five (A, D, E, S, and Y) CYP716 subgroups, with the A and Y subgroups preserving both ancestral duplicates (Fig. 5a and Supplementary Fig. 41). The five CYP716 subgroups not only neo-functionalized for oxidation and hydroxylation at different carbon positions (Fig. 5a), but also showed expression-level sub-functionalization in leaf and root tissues (Fig. 5b). More importantly, neo-functionalization and expression-level sub-functionalization of the lineage-specific proto-panaxadiol synthase (Y-subgroup) and protopanaxatriol synthase (S-subgroup) genes, together with the eudicot-common oleanolic acid synthase gene (A-subgroup) and other key genes (i.e., UGTs) of the ginsenoside biosynthesis, have facilitated the evolution of the immense diversity in structure and function of ginsenosides in *Panax* genus. Our metabolic analyses further confirmed that both dammarane-type (synthesized by S and Y subgroups) and oleanane-type (synthesized by A subgroup) ginsenosides showed different concentrations in leaf and root tissues (Fig. 5c and Supplementary Figs. 42 and 43).

### Discussion

Polyploidy is a universal phenomenon in the evolutionary history of all angiosperm plants[1,30]. Studies from across the entire phylogenetic spectrum of angiosperms have clearly illustrated the critical roles of polyploidization in genome evolution and species diversification[1,2,5]. Genome collinearity analyses have also confirmed that reshuffling of ancestral genome blocks following polyploidy resulted in great diversity of genome architecture in extant plant species[5,9,10]. In this study, focusing on the genus *Panax*, our comparative analyses revealed that while the ancestral core-eudicot genome has experienced three rounds of WGDs (γ, Pg-β and Pg-α), all these duplicated ancestral core-eudicot chromosomes (Eu1-Eu7)

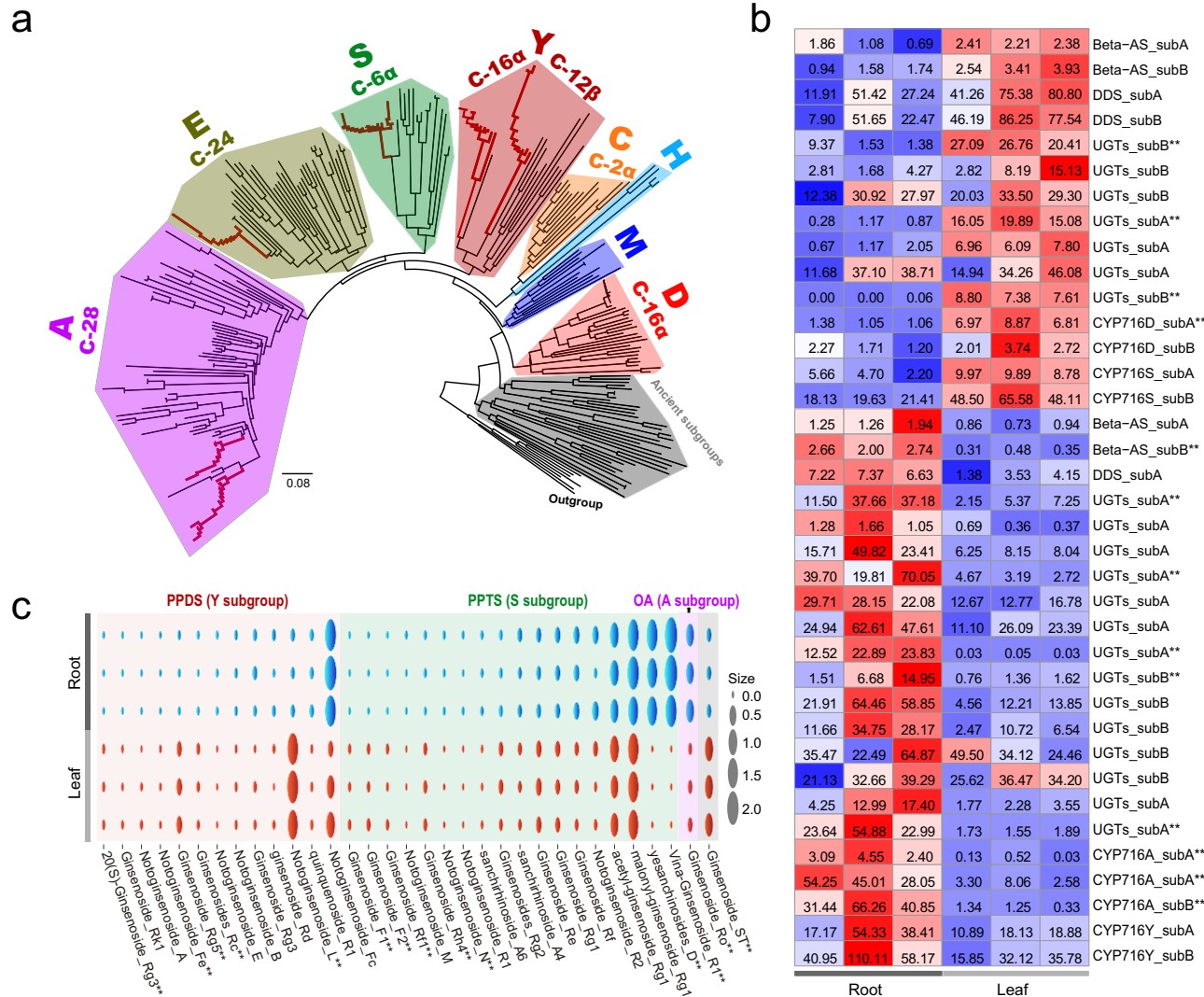

**Fig. 5 Functional diversification of the CYP716 subfamily and triterpenoid biosynthesis genes. a** Functional divergence of the subgroups in CYP716 subfamily. Each colored clade represents a subgroup that was derived from an ancestral CYP716 gene during polyploidization/diploidization processes. Branches marked with red color are *Panax* species. Oxidation and hydroxylation of the carbon positions are shown for each clade. **b** Expression divergence of the triterpenoid biosynthesis genes in root and leaf tissues. Numbers in the heatmap are the normalized gene expression levels based on transcriptome data. Name of each gene is shown on the right. The five subgroups of CYP716 were derived from paleo-polyploidizations. The "subA" and "subB" originated by Pg-α. The other major genes involved in triterpenoid biosynthesis in ginseng species are also shown in the heatmap. **c** Metabolic analyses of the triterpenoid biosynthesis in ginseng root and leaf tissues. Name of each ginsenoside is shown at the bottom. Red, green and purple colors indicate the two dammarane and one oleanane type ginsenosides, respectively. The ginsenoside ST marked with gray color is unknown. **, indicates significant difference (Wald test) between the root and leaf tissues. Exact *p* values of these comparisons are shown in the Source Data file.

are still preserved—at least to some extent—in extant *Panax* species. In particular, *Panax* species possess a relatively more conserved ancestral core-eudicot genome relative to some other extant eudicots, such as carrot and lettuce, although each of these extant species experienced additional paleo-polyploidization events after the γ hexaploidy event.

Using this paleo-genomic framework, we further examined whether this reshuffling of the ancestral core-eudicot genome has caused remodeling of chromatin topology in extant *Panax* genomes. Compared to neo-polyploid species, i.e., cotton and wheat, recent polyploidization has reshaped the chromatin topologies between the subgenomes at a small-scale level[24,25]. Our comparisons showed that chromatin topologies (A/B compartment and TAD-like structure) of the duplicated ancestral core-eudicot chromosomes (Eu1–Eu7) were extensively remodeled in extant *Panax* species. Of special significance, we show that while all seven

ancestral core-eudicot chromosomes underwent substantial changes in DNA-based genome structure, chromatin interactions within the same ancestral chromosome have been largely maintained in extant ginseng genomes after the cyclic polyploidization-diploidization processes. These findings provide a paleo-polyploidization perspective of how reshuffling of the ancestral core-eudicot genome has defined the reestablishment of chromatin topology in derived extant eudicot species.

Genome contraction after polyploidization is a common phenomenon at both the chromosomal and individual gene level. Evidence from diverse neo-polyploid species and experimental aneuploid lines have illustrated that gene dosage balance and divergence in molecular function or gene expression have determined the post-polyploidization retention of the gene duplicates[40]. Here, we show that biased genetic fractionation, together with divergent epigenetic regulation, likely played

important roles in reshaping the duplicated ancestral core-eudicot chromosomes during polyploidization and (re)diploidizations. In particular, although the reshuffling of the ancestral core-eudicot genome has resulted in genome structure diversity and epigenetic regulation divergence, genomic regions derived from the seven ancestral chromosomes likely have similar functional contributions to tissue development of extant *Panax* species.

It has been proposed that the highly dynamic nature of the genome architecture has acted as a key factor in the diversification of polyploid species[5,41,42]. We here showcase plant secondary metabolites as examples to illustrate the important roles of paleo-polyploidization/(re)diploidization processes in promoting metabolic diversity of extant eudicot plants. Our results revealed that preferential retention and regulation divergence of CYP genes have promoted the qualitative diversification of secondary metabolites in extant *Panax* species. Of significance, neo- and sub-functionalization within the CYP716 subfamily has likely resulted in immense diversity in structure and function of ginsenosides in the *Panax* genus. The plant secondary metabolites are important determinants in regulating plant development as well as the responding to various biotic and abiotic stresses[33]. Our findings suggest that different evolutionary events have played important roles in the biochemical diversity of CYPs and consequently the ecological adaptation of *Panax* species.

## Methods

### Plant materials, DNA and RNA extraction, and karyotype characterization.
To address the evolutionary reorganization of the ancestral core-eudicot genome in extant *Panax* species, we collected samples of one diploid (*P. stipuleanatus*, $2n = 2x = 24$) and three tetraploid (*P. ginseng*, *P. japonicus* and *P. quinquefolius*, $2n = 4x = 24$) species (Supplementary Note 1). Genomic DNAs was extracted from fresh mature leaves using the TianGen plant genomic DNA kit (Tianjin, China). Total RNA was extracted from leaf, stem and root tissues using the TianGen plant RNA kit. Genomic DNA and RNA for genome assembly and gene annotation were obtained from the same individual of each species. Total RNAs (mRNA, lncRNA and small RNA) for gene expression comparison were isolated from leaf, stem and root tissue of three individuals for each species using the TianGen plant RNA kit (Tianjin, China). The haploid genome size of the four species was estimated by flow cytometry with three technical replicates. Karyotypes of the four species were visualized using OLYMPUS BX53 (Olympus Corporation, Japan).

### Genome sequencing, assembly and gene annotation.
Three de novo assembly strategies were employed to reconstruct the reference genomes of the four *Panax* species (Supplementary Note 1). Briefly, short insert libraries (350 bp) of *P. stipuleanatus* and *P. japonicus* were constructed by Illumina Novaseq (Tianjin, China) and sequenced using the Illumina Novaseq platform (Illumina, USA). Then, ~20 Kb SMRTbell libraries were generated for each of the two *Panax* species and sequenced on the PacBio RSII platform (PacBio, USA). In addition, DNA fragments longer than 50 kb were used to construct a 10× Gemcode library with a Chromium instrument (10× Genomics) and sequenced using the Illumina Novaseq platform (Illumina, USA). Finally, digested genomic DNA was used to construct Hi-C library for the two species and sequenced using the Illumina Novaseq platform (Illumina, USA). In contrast, two alternative strategies using the Nanopore platform (Nanopore, UK) was employed to assemble the reference genomes of *P. ginseng* and *P. quinquefolius*, respectively. De novo assembly and genome quality control were detailed in Supplementary Information (Supplementary Note 1 and 2; Supplementary Fig. 44). Gene models were predicted based on de novo prediction, homologous identification and Unigene clusters. Repeat elements were characterized using LTR-FINDER[43] and RepeatScout[44] and annotated using RepeatMasker[45] with the parameter "-nolow -no_is -norna -engine wublast".

### Ancestral karyotype inference and ancestral gene characterization.
The ancestral core-eudicot karyotype was constructed by identification of collinear genomic blocks among the *Vitis vinifera* (grape)[18], *Daucus carota* (carrot)[46], *Lactuca sativa* (lettuce)[47] and the four *Panax* species using ColinearScan[48] using a pipeline developed in previous studies[22,49]. In brief, the grape is the most conserved genome among all extant core-eudicot plants and did not experience additional polyploidization events after its split from the common core-eudicot ancestor[18]. Carrot and lettuce are the most closely related species with assembled reference genomes to *Panax*. Protein sequences and genome annotations of the three selected species were obtained from Phytozome (https://phytozome.jgi.doe.gov/pz/portal.html). Paleo-polyploidization histories of the selected species have been well-documented; all of which have experienced the core-eudicot shared γ triplication and followed by additional species-specific duplications/triplications (i.e., Dc-α and

Dc-β)[1,12,14,20]. In addition, we inferred the karyotype of the ancestral *Panax* genome by identification of collinear genomic blocks among the four extant *Panax* species. Orthologous genes among the four *Panax* species were determined using the BLAST[50] with an *e* value cutoff of $10^{-5}$. Then, these homologous genes were used to identify collinear genes. The maximal gap length between neighboring genomic blocks was set to 50 genes[12,51–53]. Large gene families with >30 members were excluded from the genome collinearity analysis. Putative ancestral core-eudicot genes were defined as those genes shared among at least two subgenomes of the four *Panax* species. Based on the polyploidization history of the four *Panax* species, we defined the top two homologous genes (duplicated by Pg-β) as in-paralogous genes and the other two homologous genes (duplicated by γ) as out-paralogous, respectively. The order of the ancestral core-eudicot genes on genomic region of *Panax* chromosomes was determined using ColinearScan[48]. Based on the collinear genomic regions, overall genome collinearity was then visualized using WGDI (https://github.com/SunPengChuan/wgdi). In addition, we also calculated the synonymous mutation rate (Ks) between colinear homologous genes using the YN00 program in the PAML (v4.9 h) package with the Nei-Gojobori approach[54]. The median Ks value of each collinear genomic region was applied to infer the polyploidization history. In brief, we used the kernel smoothing density function to generate $K_S$ distribution curve. Then, Gaussian multipeak fitting of the curve was further generated by using the gaussian approximation function in WGDI. Orthologous gene families of the four *Panax* species were identified using OrthoFinder[55]. Then, gene family amplification and contraction analysis were performed using CAFE software with default parameters[56]. Full-length LTR retrotransposons in the *Panax* species were characterized using LTR-harvest[57] and LTR-finder[44]. Insertion time of each LTR retrotransposon family was estimated using the formula: age = K/2r, where K is the Kimura 2-parameter distance and r is the mutation rate of $1.3 \times 10^{-8}$ for the these *Panax* species[58].

### Chromatin topology and DNA methylation.
A total of 1,408,300,221,000 bp Hi-C data of the *P. stipuleanatus* (718.5× genome coverage) were obtained from the Illumina Novaseq platform (Illumina, USA). Clean short Illumina reads were mapped to the reference genome using bowtie2 (version 2.2.3)[59]. Only uniquely valid read pairs were retained for the subsequent interaction analyses. The Hi-C interaction matrix was constructed according to the pipeline developed by previous study[27,60]. Briefly, we utilized the ICE method to remove potential Hi-C data bias caused by restriction fragment length, GC content and mapping of reads. Interaction matrices at various resolutions (genome partitioned into bins of different sizes) were constructed using HiC-Pro[61] and visualized by HiC-Explorer[62]. Genome-wide Hi-C resolutions were defined as 100 Kb for compartment A/B and 20 Kb for TAD-like structure, respectively, based on interaction maps[63]. Identification of activated and inactivated compartment was performed at 100 kb resolution using the matrix2compartment module in Cworld software[64]. In brief, the expected score within the matrix was calculated using lowess smoothed average over the intra-interactions. The eigenvalues of the principal component were plotted to ascribe the bins to two types of compartments. Positive and negative eigenvalues denoted the compartment A and B, respectively. Enrichments of respective genomic compositions and epigenetic markers in each 100 kb bin were summarized and compared in terms of their compartment origins (compartment A/B), and visualized by ggplot2[65]. Topologically associated domain-like structures were identified by the insulation score method at a 20 kb resolution[66] with default parameters. Exact TAD-like structure boundaries and interior regions were specifically framed out using Hidden Markov Model[67]. Then, we converted the real TAD-like structure into a visualized format. Cytosine methylation (5mC) was calculated using Nanopolish[68]. Expression patterns of protein-coding genes were estimated using DESeq2[69].

### Expression patterns of protein-coding genes and small RNAs.
Clean non-coding RNA reads were mapped onto the reference genomes with HISAT2[70]. In parallel, the short non-coding RNA reads were also assembled by StringTie[71]. The program GffCompare (http://ccb.jhu.edu/software/stringtie/gffcompare.shtml) was used to compare the assembled transcripts to annotated protein-coding genes[72]. Long non-coding RNAs (lncRNAs) were then identified as transcripts >200 nucleotides in length which lack protein-coding potential[73]. Those lncRNAs that were expressed in only one replicate and with TPM < 1 were excluded. For the small RNAs, the clean reads were aligned and analyzed with reference genomes using ShortStack (version 3.8.3) with default settings[74]. The identified non-coding RNAs were extracted with custom Perl scripts. The number of total metabolites were estimated for the leaf and root tissue of the four species using non-targeted metabolomics. The overall quantity of the metabolites was estimated and normalized based on the total peak area in the sample. Variable importance in projection (VIP) produced by PLS-DA, ANOVA, and fold change (FC) were applied to discover the contributable variable for classification. Tissue-biased metabolites were defined according to the following parameters, including VIP > 1, *p* value < 0.05, and FC ≥ 2 or FC ≤ 0.5.

Total RNAs of three *Panax* species (*P. stipuleanatus*, *P. ginseng*, and *P. quinquefolius*) was extracted from root, stem and leaf tissues using an RNA extraction kit (Tiangen, Beijing, China) based on the manufacturer's instructions. RNA libraries were constructed by Novoseq (Tianjin, China) and sequenced using Illumina Novoseq (Illumina, CA, USA). Clean reads were aligned to the reference

genomes using HISAT with default parameters[70]. Raw mapped read counts were calculated using the prepDE.py script provided by StringTie[71]. Differences in transcription level of each gene were estimated using DESeq2[69]. Differentially expressed genes (DEGs) were defined according to the 2-FC differences ($p < 0.05$) at the transcription level between different samples. Functional annotation of the genes was performed based on the KEGG and GO databases[75]. Venn diagram were drawn with Venn Diagram[76]. Expression patterns of the protein-coding genes were estimated according to a previous study[77]. In brief, the relative expression level of a single gene was calculated based on the transcripts per million (TPM) for the three tissues within the triad as follows

$$\text{Expression(Leaf)} = \frac{TPM(Leaf)}{TPM(Leaf) + TPM(Root) + TPM(Stem)} \qquad (1)$$

$$\text{Expression(Root)} = \frac{TPM(Root)}{TPM(Leaf) + TPM(Root) + TPM(Stem)} \qquad (2)$$

$$\text{Expression(Stem)} = \frac{TPM(Stem)}{TPM(Leaf) + TPM(Root) + TPM(Stem)} \qquad (3)$$

where TPM(Leaf), TPM(Root), and TPM(Stem) represent the expression level of each gene in leaf (Eq. 1), root (Eq. 2) and stem (Eq. 3) tissues, respectively. The normalized expression value was calculated for each one of the three tissues and for the average across all expressed tissues. The values of the relative contributions of each tissue per triad were used to plot the ternary diagrams using the R package ggtern[78].

**Cytochrome P450 gene family member identification**. Orthologs of the CYP 450 superfamily in the ginseng genus and other selected species were identified using BLAST. We downloaded CYP450 members identified in *Arabidopsis* (containing 288 members, https://drnelson.uthsc.edu/cytochromeP450.html) and *P. ginseng* (containing 484 members) as references in this analysis[14]. We aligned each species' genome to the references, using blastp (-outfmt 6, -e value 1e-5, -num_threads 20, -num_alignments 100 and identity ≥ 40%), and received many potential terms. In addition, we required that the identical amino acid sites accounted for ≥ 40% (for Panax species) or ≥ 20% (for other species) of the reference gene sequence in each potential term (which will be classified into retained terms), when the CYP450 data set from *P. ginseng* (KPG) was used as reference. Finally, we compared previous terms with potential terms when the CYP450 data set from Arabidopsis was used as reference, and only overlapped terms that met the following requirements (identical amino acid sites accounted for ≥ 40% of its own gene sequence and ≥ 20% of the Arabidopsis gene sequence in each term) were kept for further analyses.

**Reporting summary**. Further information on research design is available in the Nature Research Reporting Summary linked to this article.

## Data availability
The raw total sequence reads and four genome assemblies have been deposited into the National Center for Biotechnology Information under the BioProject number PRJNA752920 with BioSample accessions SAMN20855168, SAMN20855173, SAMN20855195 and SAMN20855167. We also deposited the genome assemblies of the four *Panax* to Genome Warehouse in National Genomics Data Center[79,80], Chinese Academy of Sciences/China National Center for Bioinformation, under the project number PRJCA006678. Source data are provided with this paper.

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

## Acknowledgements
We thank James A. Birchler for his suggestions on the gene balance dosage hypothesis. This study was supported by the Natural Science Foundation of China (#31970235 to L.F.L. and 31991211 to B.L.) and the Shanghai Pujiang Program (#19PJ1401500 to L.F.L.). T.L. has been supported by a Vanier Canada Graduate Scholarship and a doctoral training fellowship from Fonds de Recherche du Québec–Santé. Y.V.d.P. acknowledges funding from the European Research Council (ERC) under the European Union's Horizon 2020 research and innovation program (No. 833522) and from Ghent University (Methusalem funding, BOF.MET.2021.0005.01).

## Author contributions
L.F.L., Y.V.P., J.F.W., and B.L. conceived this project and coordinated research activities; Z.H.W., M.R.L., S.T.L., X.Q.F., and P.J. collected and maintained the plant materials; J.Z. carried out the chromosome experiments; Z.H.W., T.L., X.F.W., M.R.L., P.J, S.T.L., and X.Q.F. conducted the comparative genomic and epigenomic analyses; Z.H.W., X.F.W., Y.V.P, J.F.W., B.L., and L.F.L. wrote the paper. All authors discussed the results and approved the paper.

## Competing interests
The authors declare no competing interests.
