## [Peer Review File · Nature Communications]

Reshuffling of the ancestral core-eudicot genome shaped chromatin topology and epigenetic modification in *Panax*Reviewers' Comments:

Reviewer #1:

Remarks to the Author:

Wang et al., presented an interesting study with several important findings by sequencing four panax genomes, tracing the structural changes following three polyploidizations, and investigating the evolutionary pattern of chromatin topologies in Panax genomes. Firstly, they found that the ancestral eudicot chromosomes are better-conserved in panax genomes relative to carrot and lettuce. The chromatin topologies of duplicated pro-chromosomes were extensively remodeled after polyploidizations, whereas the intra-chromosomal interactions of ancestral eudicots are well maintained after 100 MYs of evolution. They also proposed that biased genomic fractionation and epigenetic regulation divergence together rebalanced the duplicated ancestral genomes. Lastly, polyploidizations might result in biochemical diversity of secondary metabolites in panax. Overall, it is an interesting study with tremendous amount of new data and many novel findings.

But I still have several concerns that need to be addressed before publication.

Additional comparisons between the four newly assembled genomes are needed. For example, any differential fractionation could be identified between the three tetraploids. The authors could add the comparisons of tetraploid genomes in Fig S2 to distinguish the subgenomes in three tetraploid genomes.

I am also wondering why the genome size of the tetraploid *P. japonicas* is similar to that of the diploid *P. stipuleanatus*. I would expect some differences in the content of the repetitive sequences. But several aspects of facts need to be explored. For example, when the TEs experienced burst in these four genomes? How it related to the recent 1MA WGD? Whether the remodeled chromatin topologies, as well as the expression dynamics, are related to the repetitive sequences?

It is still not clear to me how the authors investigated the TADs evolution following the WGDs. The authors need to provide additional evidence to explain the robustness of using contact information from the modern species to infer their ancestral TAD state, even though some genomic regions well-preserved.

I have several issues with the Figure S4a. The authors need provide more details about the methods they used for identification of the rearranged chromosomes. The Vvi6, Vvi8 and Vvi13 are paralogous chromosomes generated from the gamma event. If the orthologous region of Ps4 is Vvi13, the other two homologous chromosome regions (Vvi6 and Vvi8) should be the out-paralogous regions, rather than the orthologous regions. Thereby, the authors should be explained these results carefully and clarify these identified chromosome fusions. Similar problems also exist in other chromosomes, such as Vvi10, Vvi12, and Vvi19, and the tripled chromosomes of Vvi1, Vvi14, and Vvi17. Additionally, a clear dotplots should be much helpful to illustrate these chromosomal rearrangements, and to distinguish the orthologous/out-paralogous genomic regions between genomes.

Line 151: The identified two large translocations on chromosome Ps8 and Ps9 may be from one reciprocal translocation event that results in genomic region exchanges between Ps8 and Ps9. Also, the one large inversion on Ps4 seems to be from two different chromosome rearrangement events in diploid ancestor of tetraploid Panax, because the two orthologous chromosomes of Ps4 in tetraploid Panax with different fission points. The authors could use another species as a reference genome to check whether the inversion event occurred in diploid *P. stipuleanatus* and after the P-Alpha.

Additional minor comments:

1. It is necessary to provide the Hi-C reads mapping stats during the TAD calling process.
2. Line 62 : "It is evidenced that all eudicots share an ancient WGD, usually referred to as the γ -triplication event. " It should be all extant core-eudicot. Also please check the correctness of the cited

reference.

3. Line 200 shows a range of values for the chromosomal interactions. The authors should clarify the meaning of these values for broad readers.
4. Figure 3b: the legend of "Grey color box represents the homologous genomic region that was lost in modern grape genome." is not consistent with the Figure S4b.
5. Line248-254 : "functional important genes showed nearly equal contributions to the leaf development processes, i.e., photosynthesis, kinase and synthase". Any statistical significance?
6. Line303-306 : "...mainly due to the independent retention of duplicated CYPs during the polyploidization/(re)diploidization processes... biased fractionation of the CYPs resulted in highly variable copy numbers in modern Panax genomes". Is there any specific results to support?
7. Figure S4a: (a-b) Y-axis, typo "Panax sgingenseng"

Reviewer #2:

Remarks to the Author:

In this manuscript, Wang et al., assembled chromosome-level genomes of one diploid and three tetraploid Panax species and conducted a deep comparative genomic and epigenomic analyses. Authors showed that the duplicated proto-chromosomes of the ancestral eudicot genome are well-preserved in modern Panax genomes. Authors proposed that biased genetic fractionation and epigenetic regulation divergence have together rebalanced the duplicated ancestral eudicot genome and that the polyploidization/(re)diploidization processes have generated biochemical diversity of secondary metabolites in the Panax genus. Overall, the topic is interesting and data quality is good but the part integrating chromatin organization and DNA methylation is not convincing.

1. From all the chromatin interactions described, TADs is the best characterized, as they are visually recognizable as high interaction squares along the diagonal in Hi-C matrices with at a relatively low resolution (Maass et al., 2019). They are defined as 3D structures where the chromatin regions included within a TAD interact with each other in cis with a higher frequency than with chromatin outside it. In addition to this animal TADs which are the first describe are isolated units where genes inside the same TAD are co-regulated (Dixon et al., 2012, 2016). In plants in general its was showed that genes are enriched in TAD borders whereas TE are enriched in side TAD-like structure ((Liu et al., 2017; Dong et al., 2018 ; Concia et al., 2020). Suggesting that TADs in animal and in plant are different in nature and function. In this context I suggest authors to replace TAD by TAD like structure in the text.
2. It is not clear which parameter were used to call the TAD-like structures. It is written that authors used the insulation score method. But what was the threshold used. In addition do authors see (i) that the genes are in the TAD-like borders as it is the case in other plant species? And (ii) that expressed genes are more insulated?
3. The visualization of the TAD-like structure in this article does not allow us to evaluate the TAD-like organization. For that end I suggest to authors to use a Hi-C visualization tool to generate a real Hi-C map for the figures 2B and 3C which will allow the reader to evaluate the reality of the 3D folding.
4. Authors generated DNA methylation but what is the relationship between TAD-like structure and DNA methylation. Does the DNA methylation increase within a TAD-like compare to the border?
5. One of the conclusion of the paper is that "that genetic fractionation together with divergent epigenetic regulation have rebalanced the duplicated eudicot proto-chromosomes during the repeated polyploidization/(re)diploidization processes." To my view analyzing only the DNA methylation is not enough to conclude about the impact of epigenetic regulation in this process. Authors should either do extensive analysis or tone down this conclusion.

Reviewer #3:

Remarks to the Author:

This manuscript proposes chromosome level genome sequences for four Panax species and deals with

chromosomal repatterning by rebuilding the ancestral genome structure. The manuscript includes huge data and intensive efforts to unveil chromosomal scale genome reshaping related to two independent genome duplications, following the ancient genome triplication found in the grape genome. The manuscript supports the dynamism for plant genome evolution via cyclical polyploidization/rediploidization process which contribute the genome plasticity and metabolic diversity in *Panax* species. They also show epigenetic regulation divergence and metabolic diversity caused by duplicated genes.

Basically, I agree that the manuscript reports valuable scientific finding and meaning for genome evolution of the *Panax* species of which genome was in veil due to the very complicated genome structure and limitations on genetic studies of this plant.

However, the manuscript focused on genome repatterning for the *Panax* species. Rebuilding of the ancient chromosome and the chromosome repatterning story should be based on the complete reference genome assembly given that there are no assembly errors. If the pseudo-chromosomes contain mis-assemblies, the main issue of the manuscript cannot be supported from the scientific society. I have doubts on the chromosome level genome assemblies that are not supported with enough validation for a plant with high heterozygosity and heterogeneity without clarification for the following issues.

1. *Panax* species are generally self-crossing but have high heterozygosity levels and take several years for one generation. Therefore, there are no inbred lines for each species, even for *Panax ginseng* which is maintained as relatively uniform cultivating varieties. How many plants are used for the whole genome sequencing process (Nanopore, PacBio, Illumina, Hi-C, 10x, etc.)? Even when long reads are used for assembly, heterozygosity and heterogeneous features could be problematic, making the assembly prone to mis-assembly. Also, are the samples for gDNA extraction the same as those used for RNA extraction and analyses?

2. The plant materials are not clearly described in the manuscript. Are these plants used for sequencing publicly available or have means of propagation? Because *Panax stipuleanatus* (Ps) and *P. japonicus* (Pj) are on debate for their taxonomical positions in many studies. Especially, there is almost no data for Ps. The materials section only indicates that the samples were 'collected'. Where were these samples collected, and how were the plant samples validated to be identified as the specific species? Are these plants being maintained in any institute?

3. Hi-C is a tool to increase the contiguity of scaffolds to reach the actual chromosome number. However, Hi-C itself cannot validate the assembly completeness or catch mis-assemblies of the actual chromosome-level pseudomolecule. Is there any validation data on the pseudomolecule level of each species? Perhaps genetic maps or oligo-FISH probes can be applied for this purpose.

4. How complete are the chromosome-level assemblies for the four species, and could the completeness/incompleteness possibly interfere with the conclusion drawn in this manuscript? Like the question from number 3, this study lacks pseudomolecule level validation. BUSCO values measured for the 4 species are around 93-95% which could be considered low in some cases. Moreover, *Panax quinquefolius* was assembled to be 3.57Gb in this study, which was similar to the assembled *Panax ginseng* genome size (3.66Gb). Previous studies have reported *P. quinquefolius* to have a larger genome size than *P. ginseng* via flow cytometry (You have done flow cytometry experiments provided in 'Supplementary dataset 1' which I cannot find). If this is true, your *P. quinquefolius* assembly may be lacking sequence information. Can this lead to misinterpretation of results?

5. Two *P. notoginseng* genomes, the most popular diploid *Panax* species, have already been reported in chromosome levels. However, there is no data showing synteny between Pn and Ps which could be valuable in supporting or confirming assembly completeness.

6. Figure S2 shows comparison among three genomes but do not show comparison between Pg and Pq. Any reason?

7. Illustration of synteny and more comprehensive comparison among the four species will provide more valuable data. I'd like to ask for the addition of more data for comparative genome analysis among the current four genomes instead of emphasis for the ancient genome rebuilding and repatterning.

Reply to Reviewer comments

Reviewer #1 (Remarks to the Author):

Wang *et al.*, presented an interesting study with several important findings by sequencing four *Panax* genomes, tracing the structural changes following three polyploidizations, and investigating the evolutionary pattern of chromatin topologies in *Panax* genomes. Firstly, they found that the ancestral eudicot chromosomes are better-conserved in *Panax* genomes relative to carrot and lettuce. The chromatin topologies of duplicated proto-chromosomes were extensively remodeled after polyploidizations, whereas the intra-chromosomal interactions of ancestral eudicots are well maintained after 100 MYs of evolution. They also proposed that biased genomic fractionation and epigenetic regulation divergence together rebalanced the duplicated ancestral genomes. Lastly, polyploidizations might result in biochemical diversity of secondary metabolites in *Panax*. Overall, it is an interesting study with tremendous amount of new data and many novel findings.

---Reply: We appreciate these positive comments!

But I still have several concerns that need to be addressed before publication.

Additional comparisons between the four newly assembled genomes are needed. For example, any differential fractionation could be identified between the three tetraploids. The authors could add the comparisons of tetraploid genomes in Fig S2 to distinguish the subgenomes in three tetraploid genomes.

---Reply: In the previous version of our manuscript, we identified subgenomes of the three tetraploid species according to the sequence homoeology of the two subgenomes to the diploid species (see Supplementary Table 1). In this revised version, our simulated FISH (new Supplementary Figure 31) and phylogenetic inference (new Supplementary Figure 5) confirmed that the two subgenomes show different karyotypes and phylogenetic positions. In the revised version of our manuscript, we added the comparisons of tetraploid genomes in the new Supplementary Figure 4 (Page 5, Lines 126-130).

Per your comment, we also estimated how the ancestral core-eudicot genes fractionated in the four extant *Panax* species post the Pg- β and Pg- α duplications. Our new comparisons revealed biased fractionation of these ancestral genes in the two subgenomes of the tetraploid species (new Supplementary Tables 5-9). All major changes are shown in blue color in the revised version of our manuscript (Page 6, Lines 183-190).

I am also wondering why the genome size of the tetraploid *P. japonicus* is similar to that of the diploid *P. stipuleanatus*. I would expect some differences in the content of the repetitive sequences. But several aspects of facts need to be explored. For example, when the TEs experienced burst in these four genomes? How it related to the recent 1MA WGD? Whether the remodeled chromatin topologies, as well as the expression dynamics, are related to the repetitive sequences?

---Reply: We appreciate these excellent comments! We also realized this when we know the genome features of the *P. stipuleanatus* ($2n = 2x = 24$, genome size = 1.96 Gb) and *P. japonicus* ($2n = 4x = 48$, genome size = 2.02 Gb). Therefore, we checked this with different strategies. (i) Our karyotype analyses confirmed the ploidy of the four *Panax* species (Supplementary Figure 1); (ii) Genome sizes of the four species were estimated by both genome survey and flow cytometry (Supplementary Figure 2-3); (iii) Estimates of the Ks value also identified two peaks of *P. stipuleanatus* (Pg- β and γ) and three Ks peaks (Pg- α , Pg- β and γ) in the three tetraploid species (Supplementary Figure 6). All our evidence confirmed the ploidy and genome size of the four *Panax* species.

As suggested by the reviewer, we estimated the burst times of LTRs for the four newly assembled *Panax* species and for the previously published *P. notoginseng*. Our comparisons revealed that the

distinct evolutionary history of the LTRs resulted in different content of the repeat sequences in the five *Panax* species (Supplementary Figure 12-13). We added the related content to the revised version of our manuscript (Pages 6-7, Lines 190-200 and Page 14-15, Lines 461-465).

It is still not clear to me how the authors investigated the TADs evolution following the WGDs. The authors need to provide additional evidence to explain the robustness of using contact information from the modern species to infer their ancestral TAD state, even though some genomic regions well-preserved.

---Reply: We identified genome-wide TADs on the *P. stipuleanatus* based Hi-C data. The orthologous genomic regions duplicated by distinct WGDs were identified based on genome collinearity. Then, we compared the distribution patterns of the TADs among the WGD-derived genomic regions. For example, the orthologous genomic regions on chromosomes 2 and 6 resulted from the WGD Pg- β (Figure 3c), but they possess different numbers and patterns of TADs. As the changes in TADs may affect epigenetic regulation, we then propose that different chromatin topologies between WGD-derived genomic regions may increase the genome plasticity of extant species. We did not infer the ancestral TAD state. We have now tried to better explain this in the revised version of our manuscript (Page 7-8, Lines 225-239; Page 15, Lines 480-483).

I have several issues with the Figure S4a. The authors need provide more details about the methods they used for identification of the rearranged chromosomes. The Vvi6, Vvi8 and Vvi13 are paralogous chromosomes generated from the gamma event. If the orthologous region of Ps4 is Vvi13, the other two homologous chromosome regions (Vvi6 and Vvi8) should be the out-paralogous regions, rather than the orthologous regions. Thereby, the authors should be explained these results carefully and clarify these identified chromosome fusions. Similar problems also exist in other chromosomes, such as Vvi10, Vvi12, and Vvi19, and the tripled chromosomes of Vvi1, Vvi14, and Vvi17. Additionally, a clear dotplots should be much helpful to illustrate these chromosomal rearrangements, and to distinguish the orthologous/out-paralogous genomic regions between genomes.

---Reply: We inferred karyotype evolution according to the pipeline developed by Andrew Paterson at the University of Georgia and Xiyin Wang at the North China University of Science and Technology (i.e., Wang *et al.*, 2016, *New Phytol.*). In brief, we identified the high similarity orthologous genes according to the WGD history of the *Panax* species, compared to the grape genome. We also know that, compared to the grape, the diploid *Panax* species have experienced an additional Pg- β duplication. Consequently, each of the grape genomic regions (referred to as the post- γ ancestral core-eudicot chromosome) should match with six homologous *Panax* genomic regions (duplicated by γ and Pg- β). The best two homologous genes (referred to as in-paralogous) that duplicated by Pg- β were shown with red dots. All the other four homologous genes (out-paralogous) that duplicated by γ were shown in blue dots. In the Supplementary Figure 8, both the in-paralogous (red) and out-paralogous (blue) genes were included, but we only marked the in-paralogous regions with colored boxes. Per this comment, we clarified the methodology in the Supplementary Notes (Page 5, Lines 146-156). We also updated the Supplementary Figure 8 to make it clearer.

Line 151: The identified two large translocations on chromosome Ps8 and Ps9 may be from one reciprocal translocation event that results in genomic region exchanges between Ps8 and Ps9. Also, the one large inversion on Ps4 seems to be from two different chromosome rearrangement events in diploid ancestor of tetraploid *Panax*, because the two orthologous chromosomes of Ps4 in tetraploid *Panax* with different fission points. The authors could use another species as a reference genome to check whether the inversion event occurred in diploid *P. stipuleanatus* and after the P-Alpha.

---Reply: As suggested by the reviewer, we performed collinear analyses with two additional species (*Panax notoginseng* and *Eleutherococcus senticosus*). The new genome collinearity results support

our previous inferences of the chromosomal rearrangements (Supplementary Figure 9-10). We also clarified this in the revised version of our manuscript (Pages 5-6, Lines 159-162).

Additional minor comments:

1. It is necessary to provide the Hi-C reads mapping stats during the TAD calling process.

---Reply: According to Rao *et al.* (2014), >200x genome coverage Hi-C data can provide good resolution to determine the chromatin topology. In our study, we generated 718.5x genome coverage Hi-C data for *P. stipuleanatus*. We clarified this in the revised version of our manuscript (Page 15, Line 468).

2. Line 62: "It is evidenced that all eudicots share an ancient WGD, usually referred to as the γ -triplication event. " It should be all extant core-eudicot. Also, please check the correctness of the cited reference.

---Reply: We made changes accordingly (Page 3, Lines 64-65). We also changed the "eudicot" to "core-eudicot" throughout our manuscript.

3. Line 200 shows a range of values for the chromosomal interactions. The authors should clarify the meaning of these values for broad readers.

---Reply: These values indicate the log₂-normalized frequencies of inter-chromosomal interactions based on valid Illumina read pairs. We clarified this accordingly (Page 7, Lines 225-227).

4. Figure 3b: the legend of "Grey color box represents the homologous genomic region that was lost in modern grape genome." is not consistent with the Figure S4b.

---Reply: In the Supplementary Figure 8, the collinear regions between grape and *P. stipuleanatus* were determined by the number of protein-coding genes they share. So, the size of collinear regions is not the real physical distance on the chromosome of the two species. In contrast, chromatin interactions were determined by 100-Kb sliding windows on the chromosome. This is why the chromosome 6 of *P. stipuleanatus* contains a large missing region (grey box) that show no genome collinearity with the grape genome. We have clarified this in legend of Figure 3 and Supplementary Figure 4 and Figure 15.

5. Line248-254: "functional important genes showed nearly equal contributions to the leaf development processes, i.e., photosynthesis, kinase and synthase". Any statistical significance?

---Reply: We performed t-test accordingly. All the 21 comparisons show *p* values > 0.05, except the one between Eu 1 and Eu 7 (*p* = 0.015). We added these new results to the revised version of our manuscript (Page 9, Line2 281-284) and new Supplementary Table 14.

6. Line303-306: "...mainly due to the independent retention of duplicated CYPs during the polyploidization/(re)diploidization processes... biased fractionation of the CYPs resulted in highly variable copy numbers in modern *Panax* genomes". Is there any specific results to support?

---Reply: We appreciate this comment! In the case if no biased fractionation occurred, the WGD-derived genomic regions should harbor the same numbers of CYP genes. However, as shown in the Supplementary Figure 27, members of the nine CYP clans were randomly distributed on the extant *Panax* chromosomes. In Figure 1, we inferred how the extant *Panax* chromosomes evolved from the ancestral core-eudicot genome. We therefore proposed that biased fractionation of the duplicated CYPs may lead to the different copy number of CYP clans in extant *Panax* species. We clarified this in the revised version of our manuscript (Page 11, Lines 340-345). We also added a new table (Supplementary Table 15) to show how these retained CYPs were derived from distinct duplicated modes.

7. Figure S4a: (a-b) Y-axis, typo "*Panax sginseng*"

---Reply: Corrected.

Reviewer #2 (Remarks to the Author):

In this manuscript, Wang *et al.*, assembled chromosome-level genomes of one diploid and three tetraploid *Panax* species and conducted a deep comparative genomic and epigenomic analyses. Authors showed that the duplicated proto-chromosomes of the ancestral eudicot genome are well-preserved in modern *Panax* genomes. Authors proposed that biased genetic fractionation and epigenetic regulation divergence have together rebalanced the duplicated ancestral eudicot genome and that the polyploidization/(re)diploidization processes have generated biochemical diversity of secondary metabolites in the *Panax* genus. Overall, the topic is interesting and data quality is good but the part integrating chromatin organization and DNA methylation is not convincing.

---Reply: We appreciate these comments. Please see our point-to-point responses below.

1. From all the chromatin interactions described, TADs is the best characterized, as they are visually recognizable as high interaction squares along the diagonal in Hi-C matrices with at a relatively low resolution (Maass *et al.*, 2019). They are defined as 3D structures where the chromatin regions included within a TAD interact with each other in cis with a higher frequency than with chromatin outside it. In addition to this animal TADs which are the first describe are isolated units where genes inside the same TAD are co-regulated (Dixon *et al.*, 2012, 2016). In plants in general its was showed that genes are enriched in TAD borders whereas TE are enriched in side TAD-like structure (Liu *et al.*, 2017; Dong *et al.*, 2018; Concia *et al.*, 2020). Suggesting that TADs in animal and in plant are different in nature and function. In this context I suggest authors to replace TAD by TAD like structure in the text.

---Reply: Many thanks for this summary of the TADs. As requested by the reviewer, we replaced “TAD” with “TAD-like structure” throughout our revised manuscript.

2. It is not clear which parameter were used to call the TAD-like structures. It is written that authors used the insulation score method. But what was the threshold used. In addition, do authors see (i) that the genes are in the TAD-like borders as it is the case in other plant species? And (ii) that expressed genes are more insulated?

---Reply: We employed the default parameters to determine the TAD-like structures. We clarified this in the revised version of our manuscript (Page 15, Lines 480-483). Per this comment, we also compared the expression level and cytosine methylation of genes in the TAD-like structures and in the TAD borders (Page 8, Lines 236-239). All new results are shown in the new Supplementary Figure 17.

3. The visualization of the TAD-like structure in this article does not allow us to evaluate the TAD-like organization. For that end I suggest to authors to use a Hi-C visualization tool to generate a real Hi-C map for the figures 2B and 3C which will allow the reader to evaluate the reality of the 3D folding.

---Reply: As requested by the reviewer, we now regenerated the Hi-C map with the same data matrix. The real interaction map was visualized for each chromosome at 20-Kb resolution (new Supplementary Figure 16). In Figures 2B and 3C, we only converted the real TAD-like structure to a visualized triangle shape, so that the differences in TAD distribution among WGD-derived genomic regions can be shown clearer. But the size and distribution of the TAD-like structure are the same between the real TAD map (in Supplementary Figure 16) and transformed triangle shape (in Figure 2 and 3). Per this comment, we clarified this in the revised version of our manuscript (Page 8, Lines 234-236; Page 15, Lines 480-483).

4. Authors generated DNA methylation but what is the relationship between TAD-like structure and DNA methylation. Does the DNA methylation increase within a TAD-like compare to the border?

---Reply: For the revised version of our manuscript, we estimated the correlation between DNA methylation and TAD-like structures. As shown in the new Supplementary Figure 17, DNA methylation increases within the TAD-like structure. We added this observation to the revised version of our manuscript (Page 8, Lines 236-239).

5. One of the conclusions of the paper is that “that genetic fractionation together with divergent epigenetic regulation have rebalanced the duplicated eudicot proto-chromosomes during the repeated polyploidization/(re)diploidization processes.” To my view analyzing only the DNA methylation is not enough to conclude about the impact of epigenetic regulation in this process. Authors should either do extensive analysis or tone down this conclusion.

---Reply: We have toned down this conclusion in the revised version of our manuscript (Pages 12-13, Lines 397-407).

Reviewer #3 (Remarks to the Author):

This manuscript proposes chromosome level genome sequences for four *Panax* species and deals with chromosomal repatterning by rebuilding the ancestral genome structure. The manuscript includes huge data and intensive efforts to unveil chromosomal scale genome reshaping related to two independent genome duplications, following the ancient genome triplication found in the grape genome. The manuscript supports the dynamism for plant genome evolution via cyclical polyploidization/rediploidization process which contribute the genome plasticity and metabolic diversity in *Panax* species. They also show epigenetic regulation divergence and metabolic diversity caused by duplicated genes. Basically, I agree that the manuscript reports valuable scientific finding and meaning for genome evolution of the *Panax* species of which genome was in veil due to the very complicated genome structure and limitations on genetic studies of this plant.

---Reply: We appreciate these constructive comments.

However, the manuscript focused on genome repatterning for the *Panax* species. Rebuilding of the ancient chromosome and the chromosome repatterning story should be based on the complete reference genome assembly given that there are no assembly errors. If the pseudo-chromosomes contain mis-assemblies, the main issue of the manuscript cannot be supported from the scientific society. I have doubts on the chromosome level genome assemblies that are not supported with enough validation for a plant with high heterozygosity and heterogeneity without clarification for the following issues.

---Reply: We agree with the reviewer that the quality of the assembled genomes can affect the reconstruction of ancient or ancestral karyotypes. Therefore, reference genomes of the four *Panax* species were assembled independently by Novogene (Tianjin, China) and Biomarker (Beijing, China). Both companies have experienced bioinformatics teams and assembled and published many high-quality genomes in the top tier journals. In particular, the four *Panax* genomes were assembled using different sequencing platforms and distinct assembly strategies (see details in Supplementary Notes). We also checked the quality of the four genome assemblies based on DNA-sequence, protein-coding gene completeness and repeat sequence contiguity. All our assessments confirmed the high quality of our assembled genomes.

We also agree with the reviewer that we needed to provide more evidence to support our inferences. Therefore, we reexamined the quality of our genome assemblies with three different strategies: (1) we performed genome collinearity analyses between the four *Panax* species with previously published genomes of *Panax notoginseng* and *Eleutherococcus senticosus*; (2) we reconstructed the phylogeny of the five available *Panax* genomes and six outgroup species; (3) we simulated the karyotype of the five *Panax* species with previously developed FISH probes. All the new results again confirmed the high quality of our genome assemblies.

In addition, the methodology we used to infer karyotype evolution is now widely employed to reconstruct the ancestral karyotype of many different plant species (Wang *et al.*, 2016, *New Phyto.*; Guo *et al.*, 2019, *Genome Res.*; Zhuang *et al.*, 2019, *Nat. Genet.*; Song *et al.*, 2020, *Plant Biotech. J.*). The karyotypes of the *Panax* species were reconstructed based on independent inferences. We believe that the ancestral karyotype inferred is highly reliable. Please see the point-to-point responses below.

1. *Panax* species are generally self-crossing but have high heterozygosity levels and take several years for one generation. Therefore, there are no inbred lines for each species, even for *Panax ginseng* which is maintained as relatively uniform cultivating varieties. How many plants are used for the whole genome sequencing process (Nanopore, PacBio, Illumina, Hi-C, 10x, etc.)? Even when long reads are used for assembly, heterozygosity and heterogeneous features could be problematic, making the assembly prone to mis-assembly. Also, are the samples for gDNA extraction the same as those used for RNA extraction and analyses?

---Reply: We agree with the reviewer that it is really hard to get inbred lines for the four *Panax* species. However, the third-generation sequencing platforms (*i.e.*, Nanopore and PacBio) are now widely used to assemble reference genome of natural plant species. In our study, the two species, *P. stipuleanatus* and *P. japonicus*, were assembled at Novogene (Tianjin, China) using “PacBio+10xgenomics+Hi-C+Illumina”. The other two species, *P. ginseng* and *P. quinquefolius*, were assembled at Biomarker (Beijing, China) using “Nanopore+Hi-C+Illumina”. As we mentioned above, the two companies have professional bioinformatics teams to assemble high quality genomes. We also checked the quality of the four *Panax* genomes with different strategies (see details in Supplementary Notes, Page 2-4, Lines 40-108). We believe the inferred ancestral karyotypes of the four *Panax* species are reliable.

For the genome assembly and gene annotation, all DNA and RNA were extracted from the same individual. We have clarified all this in the main text of our revised paper (Page 13, Lines 420-425) and the revised Supplementary Notes (Page 2, Lines 35-39).

2. The plant materials are not clearly described in the manuscript. Are these plants used for sequencing publicly available or have means of propagation? Because *Panax stipuleanatus* (Ps) and *P. japonicus* (Pj) are on debate for their taxonomical positions in many studies. Especially, there is almost no data for Ps. The materials section only indicates that the samples were ‘collected’. Where were these samples collected, and how were the plant samples validated to be identified as the specific species? Are these plants being maintained in any institute?

---Reply: Samples of *P. ginseng* and *P. quinquefolius* were collected from the Jilin Province in China. Samples of *P. stipuleanatus* and *P. japonicus* were provided by our collaborators Yue-Zhi Pan from the Yunnan Province in China and Ritsuko Kitagawa from Japan. As all the four *Panax* species are small herbs, these samples used for genome assembly are not available now. But we have their sibling plants, which are grown in the greenhouse at Fudan University in Shanghai (China).

Regarding the taxonomical positions of the four *Panax* species, Prof. Jun Wen at the Smithsonian National Museum of Natural History is the expert. She classified the genus *Panax* as three tetraploids (*P. ginseng*, *P. quinquefolius* and *P. japonicus*), four diploids (*P. notoginseng*, *P. pseudoginseng*, *P. stipuleanatus* and *P. trifolius*), and one species complex (Wen and Zimmer, 1996, *Mol. Phylogenet. Evol.*; Lee and Wen, 2004, *Mol. Phylogenet. Evol.*; Zuo *et al.*, 2015, *J. Syst. Evol.*; Zuo *et al.*, 2017, *Mol. Phylogenet. Evol.*). We also cooperated with Prof. Jun Wen and other colleagues to reconstruct the phylogenetic trees of these *Panax* species (Shi *et al.*, 2015, *BMC Plant Biol.*; Jiang *et al.*, 2018, *Front. Plant Sci.*). Taxonomic positions of the four *Panax* species used in this study have been confirmed in our previous studies. In this study, we further examined the karyotypes and phylogenetic relationships of the four species (Supplementary Figure 1 and 5). All our evidence confirmed the taxonomic positions of the four species. We have clarified this the revised version of our manuscript (Main text, Page 5, Lines 126-130 and Supplementary Notes, Page 2, Lines 35-39).

3. Hi-C is a tool to increase the contiguity of scaffolds to reach the actual chromosome number. However, Hi-C itself cannot validate the assembly completeness or catch mis-assemblies of the actual chromosome-level pseudomolecule. Is there any validation data on the pseudomolecule level of each species? Perhaps genetic maps or oligo-FISH probes can be applied for this purpose.

---Reply: We agree with the reviewer. Prof. Tae-Jin Yang at Seoul National University did excellent FISH analyses for the *Panax* species (Chio *et al.*, 2014; Lee *et al.*, 2017; Shim *et al.*, 2021). In particular, his group developed a probe *PgDel2* that can separate the tetraploid species as two subgenomes. As requested by the reviewer, we simulated the karyotypes of five *Panax* species using the available FISH probes. As shown in Supplementary Figure 31, our simulated karyotypes are highly similar to the real FISH karyotypes. More importantly, subgenome B of the three tetraploid species and the two diploid species show stronger *PgDel2* signal compared to subgenome A. These results confirmed the quality of our genome assemblies. We have added the related content to the revised version of our manuscript (Supplementary Notes, Page 4, Lines 102-107).

4. How complete are the chromosome-level assemblies for the four species, and could the completeness/incompleteness possibly interfere with the conclusion drawn in this manuscript? Like the question from number 3, this study lacks pseudomolecule level validation. BUSCO values measured for the 4 species are around 93-95% which could be considered low in some cases. Moreover, *Panax quinquefolius* was assembled to be 3.57Gb in this study, which was similar to the assembled *Panax ginseng* genome size (3.66Gb). Previous studies have reported *P. quinquefolius* to have a larger genome size than *P. ginseng* via flow cytometry (You have done flow cytometry experiments provided in 'Supplementary dataset 1' which I cannot find). If this is true, your *P. quinquefolius* assembly may be lacking sequence information. Can this lead to misinterpretation of results?

---Reply: Quality control of the four assembled *Panax* genomes was performed for the DNA-based pseudomolecules, protein-coding genes and repeat sequences, respectively. Please see our detailed explanations below.

(1) Our results showed that 98.36-99.67% of the Illumina short reads can be mapped onto the assembled genomes, indicating that the genome assemblies include almost all of the genomic regions;

(2) All the four *Panax* species show high BUSCO values (93.10-95.14%), indicating the high completeness of the protein-coding genes. We also checked the BUSCO values of six previously published *Panax* genomes. Right now, four versions of a reference genome are available for *P. notoginseng*. Of the two contig-level *P. notoginseng* versions, one is only 82.4% (Chen *et al.*, 2017, *Mol. Plant*) while the other does not provide this value (Zhang *et al.*, 2017, *Mol. Plant*). Likewise, BUSCO values of the two chromosome-level *P. notoginseng* versions are 90.6% (Fan *et al.*, 2021, *iScience*) and 96.6% (Jiang *et al.*, 2021, *Plant Commu.*), respectively.

For the two contig-level versions of *P. ginseng*, the BUSCO values are 91.88% (Xu *et al.*, 2017, *GigaScience*) and 93.00% (Kim *et al.*, 2018, *Plant Bio. J.*), respectively. It is notable that the version of *P. ginseng* assembled by Kim *et al.* (2018) is much better than the other one assembled by Xu *et al.* (2017). In particular, all the detailed information of the genome version (Kim *et al.*, 2018) is available on the website (<http://ginsengdb.snu.ac.kr/index.php>), which provides a very convenient way to share the valuable genome data. As shown in our Supplementary Figure 26, we retrieved the *CYP450* genes from Kim *et al.* (2018). Then, we used these *CYP450* genes searched against our four newly assembled genomes. All the *CYP450* genes identified in the Kim *et al.* (2018) genome, were found in our *P. ginseng* genome. All the above evidence confirms the high gene completeness of our genome assemblies.

(3) The LAI assessment (LAI = 7.13-16.24) of the four species also revealed high genome contiguity of the repeat sequence. In addition, we also compared the LAI values of our *Panax* genomes with previously published genomes of the model species *Arabidopsis thaliana* and *Vitis*

vinifera. Although the four *Panax* species have much larger genomes, our genome assemblies have high LAI values that are comparable to the two model species. Together, these results again confirm the high genome completeness and contiguity of our assembled genomes. Please see the detailed explanation in Supplementary Notes (Pages 2-4, Lines 40-108).

Regarding the genome sizes of the four *Panax* species, we performed both flow cytometry and genome survey. The genome assemblies of the four species are close to those estimated by genome survey (Table 1). Our quality control also confirmed the high genome completeness and contiguity of the four assembled genomes (see details above). As requested by the reviewer, we redid the genome survey for *P. ginseng* and *P. quinquefolius* with different methods. The re-estimated genome sizes of the two species are highly similar to what has been estimated previously (see Table 1). However, we agree with the reviewer that the genome size of *P. quinquefolius* estimated by flow cytometry (4.14 Gb) is larger than those of estimated by genome survey (3.60 Gb) (Supplementary Figure 2-3) and the Hi-C version assembled genome (3.57 Gb) (Table 1). In fact, similar phenomena are also observed in genome assemblies of *P. ginseng* (varying from 2.98 Gb to 3.43 Gb) and *P. notoginseng* (varying from 1.85 Gb to 2.66 Gb). Taking the *P. ginseng* genome as an example, while the version assembled by Kim *et al.* (2018) (2.98 Gb) is apparently smaller than the other one assembled by Xu *et al.* (2017) (3.43 Gb), as well as the one estimated by genome survey (3.41 Gb) and flow cytometry (3.34 Gb). Our genome-wide comparisons confirmed that the Kim *et al.* (2018) version is much better than the version presented by Xu *et al.* (2017). We think that the small genome size presented by Kim *et al.* (2018) is possible due to the failure of the assembly of repeat sequences. The same is probably true for our *P. quinquefolius* genome. In any case, we believe that this would not affect the inference of karyotype evolution of our study, because our conclusions are based on independent inferences of different species based on protein-coding genes (not repeat sequences). As requested by the reviewer, we now provide more details leading to a three pages explanation in the Supplementary Notes (Pages 2-4, Lines 40-110).

5. Two *P. notoginseng* genomes, the most popular diploid *Panax* species, have already been reported in chromosome levels. However, there is no data showing synteny between Pn and Ps which could be valuable in supporting or confirming assembly completeness.

---Reply: We agree with the reviewer and have now also considered genome collinearity between our genomes and the previously published *P. notoginseng* genome (new Supplementary Figure 9). We also described these new results in the revised version of our manuscript (Main text, Pages 5-6, Lines 159-162 and Supplementary Notes, Page 3, Lines 86-97). These new collinearity analyses again confirmed the high quality of our genomes.

6. Figure S2 shows comparison among three genomes but do not show comparison between Pg and Pq. Any reason?

---Reply: In the previous version, we intended to show the differences in genome collinearity between the diploid and tetraploid species. We have now performed additional genome collinearity analyses among the three tetraploid species (new Supplementary Figure 4).

7. Illustration of synteny and more comprehensive comparison among the four species will provide more valuable data. I'd like to ask for the addition of more data for comparative genome analysis among the current four genomes instead of emphasis for the ancient genome rebuilding and repatterning.

---Reply: Our idea was to address how the repeated polyploidizations-diploidizations contributed to genome structure diversity and the metabolic diversity of *Panax* species. Our study provides novel insights on the evolutionary roles of ancient WGDs. As requested by the reviewer, we further performed the genome-wide comparisons among the four *Panax* species, particularly the biased

genetic fractionation among the three tetraploid species (Page 6, Lines 183-190). We added all the new results in Supplementary Tables 5-9.

Reviewers' Comments:

Reviewer #1:

Remarks to the Author:

This is the manuscript I reviewed a while ago. After carefully reading the revised version and the response letter, I think the authors have done a pretty good job in addressing my previous concerns and questions. I have no further comments.

Reviewer #2:

Remarks to the Author:

The authors fully addressed all concerns.

Reviewer #3:

Remarks to the Author:

Overall, the manuscript was fully updated and well-shaped. However, I have some concerns on a few issues.

1. *P. japonicus* (Pj) was known to be diploids or tetraploids depending on collection sites. Jun Wen's papers and other papers also represented confusion on the phylogenetic position and chromosome numbers for Pj (Yang 1981, *J. Syst. Evol.*; Wen and Zimmer, 1996, *Mol. Phylogenet. Evol.*; Lee and Wen, 2004, *Mol. Phylogenet. Evol.*; Zuo et al., 2015, *J. Syst. Evol.*; Zuo et al., 2017, *Mol. Phylogenet. Evol.*) Since this genome sequence might become a reference genome sequence for the society, it is important to clarify the origin of the four species.
2. Figure S2 displays two different formats for the K-mer analysis. Company names cannot represent the proper protocols for the analysis. The figures should be updated by one standardized platform with the same protocol and criteria.
3. Genome assemblies of the four species relied on two independent companies. The two companies probably have different assembly pipelines and experiences. As the authors mentioned, both companies are really professional for genome assemblies, but there are always assembly errors that could appear. I am not really convinced that all four do not really contain mis-assemblies. The manuscript handles mainly on chromosome level reshuffling; however, that might not be supported very well if there were mis-assemblies.

REPLY TO THE REVIEWER COMMENTS

Reviewer #1 (Remarks to the Author):

This is the manuscript I reviewed a while ago. After carefully reading the revised version and the response letter, I think the authors have done a pretty good job in addressing my previous concerns and questions. I have no further comments.

---Reply: We appreciate these positive comments!

Reviewer #2 (Remarks to the Author):

The authors fully addressed all concerns.

---Reply: We appreciated all comments, which largely improved our manuscript!

Reviewer #3 (Remarks to the Author):

Overall, the manuscript was fully updated and well-shaped. However, I have some concerns on a few issues.

1. *P. japonicus* (Pj) was known to be diploids or tetraploids depending on collection sites. Jun Wen's papers and other papers also represented confusion on the phylogenetic position and chromosome numbers for Pj (Yang 1981, *J. Syst. Evol.*; Wen and Zimmer, 1996, *Mol. Phylogenet. Evol.*; Lee and Wen, 2004, *Mol. Phylogenet. Evol.*; Zuo *et al.*, 2015, *J. Syst. Evol.*; Zuo *et al.*, 2017, *Mol. Phylogenet. Evol.*) Since this genome sequence might become a reference genome sequence for the society, it is important to clarify the origin of the four species.

---Reply: This is good suggestion. As the reviewer mentioned, traditional classification of *Panax japonicus* based on morphological traits defined all the subspecies and varieties naturally distributed in Japan and China as a single species. However, recent studies based on morphology, phylogeny and karyotype treated the Japanese subspecies as *P. japonicus* ($2n=4x=48$). All other Chinese subspecies or varieties were redefined as *Panax bipinnatifidus* species complex (Zuo *et al.*, 2015, *J. Syst. Evol.*; Shi *et al.*, 2015; *BMC Plant Biol.*; Zuo *et al.*, 2017, *Mol. Phylogenet. Evol.*; Zhou *et al.*, 2020, *Mol. Phylogenet. Evol.*). In our study, the specimen was collected from Hokkaido, Japan (see Supplementary Notes, Page 2, Lines 36-37). We have further clarified this in the revised version of our manuscript (Supplementary Notes, Page 2, Lines 39-44).

2. Figure S2 displays two different formats for the K-mer analysis. Company names cannot represent the proper protocols for the analysis. The figures should be updated by one standardized platform with the same protocol and criteria.

---Reply: Genome survey of the four *Panax* species was performed with the same pipeline. Per this comment, we updated the Figure S2. We also revised our manuscript based on this new figure (Supplementary Notes, Page 2, Line 48).

3. Genome assemblies of the four species relied on two independent companies. The two companies probably have different assembly pipelines and experiences. As the authors mentioned, both companies are really professional for genome assemblies, but there are always assembly errors that could appear. I am not really convinced that all four do not really contain mis-assemblies. The manuscript handles mainly on chromosome level reshuffling; however, that might not be supported very well if there were mis-assemblies.

---Reply: We appreciate this comment!

Firstly, we admit that our genome assemblies of the four *Panax* species are not perfect. As the reviewer indicated, it is almost impossible to exclude all mis-assemblies with current sequencing technologies and assembly pipelines. Actually, this is true for the large majority of published (plant) reference genomes. For example, although reference genomes of the model species rice and

Arabidopsis have been updated several times in the past decades, gap-free genome assemblies of the two species are only recently generated (Song *et al.*, 2021, *Mol. Plant*; Wang *et al.*, 2021, *Genom. Proteom. Bioinf.*). Still, 'gap-free reference genome' does not mean there are definitely no mis-assemblies. With this reasoning, we think that our genome assemblies are of similar quality.

Secondly, we agree with the reviewer that genome quality is very important to infer a reliable karyotype. To this end, we evaluated the quality of the four genome assemblies based on DNA-sequence, protein-coding gene completeness and repeat sequence contiguity (see in Supplementary Notes, Page 3, Lines 81-102). As requested by the reviewer, we also performed additional validations for the four *Panax* genomes, such as genome collinearity, simulated karyotype and phylogenetic inference (see in Supplementary Notes, Page 4, Lines 103-112). Again, we admit that our genome assemblies are not perfect, although we employed distinct strategies to check the quality of the four *Panax* genomes. As far as we know, we did all we could to improve the quality of our genome assemblies with the current sequencing technologies and assembly strategies. So, we believe that the quality of the four genome assemblies are good enough to reconstruct a reliable ancestral karyotype.

Finally, the methodology we used to infer karyotype evolution is now widely employed to reconstruct the ancestral karyotype of many different plant species (*i.e.*, Guo *et al.*, 2019, *Genome Res.*; Zhuang *et al.*, 2019, *Nat. Genet.*; Badouin *et al.*, 2017, *Nature*; Kreplak *et al.*, 2019, *Nat. Genet.*). Inference of the ancestral karyotype mainly relied on the identification of homologous genomic regions (not the whole chromosome) between the two selected genomes. Then, in-paralogous and out-paralogous genomic regions are determined by the synonymous substitution rate among paralogous genes. This is why the scaffold-level genome of the *Amborella trichopoda* has been widely used as a reference to infer the karyotype evolution of the other chromosome-level genomes (*i.e.*, Amborella Genome Project, 2013, *Science*; Murat *et al.*, 2017, *Nat. Genet.*).

However, as the reviewer mentioned, it is really hard to exclude all the mis-assemblies from the assembled genomes. With this reasoning, to reduce the mis-assemblies resulted from the assembly pipelines, we asked two independent companies to assemble the reference genomes of the four species with different strategies. To this end, the possibility that the mis-assemblies occur in the same physical position of the chromosome of the seven extant *Panax* genomes (one diploid + six tetraploid subgenomes) should be low. In addition, to reconstruct a reliable ancestral karyotype, only those chromosomal fusions/fissions that are commonly identified in all the seven extant *Panax* genomes are employed to infer the karyotype evolution of ancestral core-eudicot genome. Together, we have tried our best to minimize the errors caused by mis-assemblies.

We feel the reviewer's comments are very relevant for the current study and have therefore clarified this in the revised version of our manuscript (Supplementary Notes, Page 4, Lines 112-114).

Reviewers' Comments:

Reviewer #3:

Remarks to the Author:

The taxonomical issue for *P. japonicus* is still complicated and on the dispute. However, the authors tried to clear every issue and did their best efforts for clarification of the most issues. I have no more comments on the last revision.

Reviewer #3 (Remarks to the Author):

The taxonomical issue for *P. japonicus* is still complicated and on the dispute. However, the authors tried to clear every issue and did their best efforts for clarification of the most issues. I have no more comments on the last revision.

---Reply: We appreciate the reviewer's comments, which have largely improved our manuscript. Per the taxonomical position of *P. japonicus*, we agree with the reviewer, as we explained previously, that traditional classification of this species is complicated. However, the latest classification system of the genus *Panax* clearly defined the Chinese subspecies and varieties as *Panax bipinnatifidus* species complex. We also checked the karyotype, genome size and genome feature (Ks distribution). In any case, the specimen used in our study is a tetraploid species, which is phylogenetically close to the other two tetraploids *Panax ginseng* and *Panax quinquefolius*.